# The *Vibrio vulnificus* stressosome is an oxygen-sensor involved in regulating iron metabolism

Veronika Heinz [1,10], Wenke Jäckel[2,10], Susann Kaltwasser [3,10], Laura Cutugno [4], Patricia Bedrunka[5], Anica Graf[2], Alexander Reder[6], Stephan Michalik [6], Vishnu M. Dhople[6], M. Gregor Madej [1], Maria Conway[5], Marcus Lechner[5], Katharina Riedel[2], Gert Bange [5], Aoife Boyd[4], Uwe Völker [6], Richard J. Lewis[7,9], Jon Marles-Wright [7,8], Christine Ziegler [1✉] & Jan Pané-Farré [5✉]

Stressosomes are stress-sensing protein complexes widely conserved among bacteria. Although a role in the regulation of the general stress response is well documented in Gram-positive bacteria, the activating signals are still unclear, and little is known about the physiological function of stressosomes in the Gram-negative bacteria. Here we investigated the stressosome of the Gram-negative marine pathogen *Vibrio vulnificus*. We demonstrate that it senses oxygen and identified its role in modulating iron-metabolism. We determined a cryo-electron microscopy structure of the *Vv*RsbR:*Vv*RsbS stressosome complex, the first solved from a Gram-negative bacterium. The structure points to a variation in the *Vv*RsbR and *Vv*RsbS stoichiometry and a symmetry breach in the oxygen sensing domain of *Vv*RsbR, suggesting how signal-sensing elicits a stress response. The findings provide a link between ligand-dependent signaling and an output – regulation of iron metabolism - for a stressosome complex.

[1] Department of Biophysics II/Structural Biology, University of Regensburg, 93053 Regensburg, Germany. [2] Department of Microbial Physiology and Molecular Biology, University of Greifswald, 17487 Greifswald, Germany. [3] Max Planck Institute of Biophysics, Max-von-Laue-Strasse 3, 60438 Frankfurt am Main, Germany. [4] Discipline of Microbiology & Centre for One Health, School of Natural Sciences, Molecular Pathogenesis Research Group, National University of Ireland Galway, Galway, Ireland. [5] Center for Synthetic Microbiology (SYNMIKRO) & Department of Chemistry, Philipps-University Marburg, Karl-von-Frisch-Strasse 14, 35043 Marburg, Germany. [6] Interfaculty Institute for Genetics and Functional Genomics, University Medicine Greifswald, Greifswald, Germany. [7] Biosciences Institute, Newcastle University, Newcastle upon Tyne NE2 4HH, UK. [8] School of Natural and Environmental Sciences, Newcastle University, Newcastle upon Tyne NE1 7RU, UK. [9] Present address: The Royal Society for the Protection of Birds, The Lodge, Potton Road, Sandy, Bedfordshire SG19 2DL, UK. [10] These authors contributed equally: Veronika Heinz, Wenke Jäckel, Susann Kaltwasser. ✉email: christine.ziegler@biologie.uni-regensburg.de; jan.panefarre@chemie.uni-marburg.de

Bacteria continuously respond to environmental changes, stress, and, in the case of pathogens, the host. In many bacteria, cellular stress signals are integrated by the stressosome, a high molecular weight signaling complex, first described in the Gram-positive soil bacterium *Bacillus subtilis*[1–3] (Supplementary Fig. 1). The *B. subtilis* stressosome consists of multiple copies of two major components: a small single STAS (sulfate transporter anti sigma factor antagonist) domain protein, RsbS, and at least one out of five RsbR paralogs, which have an N-terminal sensory domain and a C-terminal STAS domain that interacts with RsbS[4–6] to form the stressosome core. The activating signals for the different RsbR paralogs have not yet been identified, with the exception of YtvA, which responds to blue light[7–10]. In the non-stressed state, the protein kinase RsbT is bound to the stressosome core[5]. Upon stress perception by the RsbR sensory domains, RsbT phosphorylates both RsbR and RsbS and is released from the stressosome[11–15]. RsbT then initiates a signaling cascade involving proteins RsbU, RsbV, and RsbW, resulting in sigma factor B (SigB)-dependent up-regulation of the transcription of stress-responsive genes[11,16–20]. Dephosphorylation of RsbS and RsbR by the phosphatase RsbX allows for re-binding of RsbT, and as such 'resets' the sensing state of the stressosome[11,13]. In cryo-electron microscopy (cryo-EM) reconstructions of the *B. subtilis* and the *Listeria monocytogenes* stressosomes, the RsbR:RsbS STAS domain core adopts a pseudo-icosahedral scaffold with the N-terminal sensory domains of RsbR projecting from this as turret-like extensions[5,6,21]. All of these reconstructions were obtained from recombinant proteins, with different methods of reconstituting the stressosome complexes; as a result of this, they show considerable differences in the arrangement of the RsbR proteins in the complex.

Phylogenetic analysis suggests that the extended eight-gene stressosome operon (*rsbRSTUVWsigBrsbX*) encoding the three stressosome proteins (RsbR, RsbS and RsbT) the stressosome feedback-phosphatase (RsbX) and the SigB activation cascade (RsbU, RsbV and RsbW) is not common in nature, and seems restricted to the order Bacillales[22,23]. Instead, a four-gene module (*rsbRSTX*) encoding a minimal stressosome but lacking the *sigB* gene and the SigB activation cascade (*rsbU, rsbV* and *rsbW*), is widely conserved in diverse bacteria including Cyanobacteria, Bacteroidetes, Proteobacteria, Deinococci and even some archaeal species. The aim of this study was to explore the structure and function of the thus far neglected stressosomes of Gram-negative bacteria. An example for the impact of the four-gene *rsbRSTX* operon was recently described for the γ-Proteobacterium *Vibrio vulnificus*[24], a pathogen associated with high mortality in patients developing infection from sea water through open wounds and by ingestion of contaminated seafood[25]. Whole genome sequencing has shown that the stressosome gene cluster is particularly well conserved in clinical *V. vulnificus* isolates (Biotype 1) and less frequently found in environmental strains (Biotype 2), suggesting that the stressosome may play a role during infection[24,26]. Furthermore, the *rsbRSTX* locus is transcribed during periods of reduced oxygen availability in sea water, showing that the stressosome operon is active in the natural habitat of this pathogen[27]. The *rsbRSTX* gene cluster is also conserved in other important pathogenic *Vibrio* species including *Vibrio corallilyticus*, causing global coral bleaching[28], *Vibrio nigripulchritudo*, an emerging pathogen of farmed shrimp[29], and *Vibrio brasiliensis*[30] that was isolated from a marine aquaculture environment[31].

Herein we show that starvation in *V. vulnificus* correlates with the expression of stressosome components, and that starvation and oxygen-limitation trigger alterations in the proteome. We also show that *Vv*RsbR and *Vv*RsbS interact in vivo and demonstrate that in vitro, they assemble into a stressosome-like complex that is sensitive to O₂ levels. Finally, we present the first cryo-EM reconstruction of a stressosome complex from a Gram-negative bacterium, assembled from *Vv*RsbR and *Vv*RsbS proteins heterologously co-expressed in *E. coli* and purified under aerobic conditions. The *Vv*RsbR:*Vv*RsbS assembly suggests that *Vv*RsbR:*Vv*RsbS stoichiometries vary, which could serve to adjust the activation threshold upon stress sensing, by changing the surface available for *Vv*RsbT binding. Together, our data link the iron-heme-dependent response of a minimal stressosome to iron-metabolism of an important pathogen, establishing *Vibrio*-type stressosomes as a model system to study stressosome biology from signal sensing to regulatory output.

## Results

### Starvation-dependent expression of the minimal stressosome.
We initially wanted to discover the conditions in which the *V. vulnificus* stressosome is produced. Western-blotting analysis, employing antibodies directed against the *Vv*RsbR protein, clearly showed that this stressosome protein accumulates in defined chemical media in stationary phase cells of *V. vulnificus*, likely induced by the exhaustion of glucose in the growth medium (Fig. 1a, b, c). This is in contrast to *B. subtilis*, where stressosome levels remain relatively constant during growth and stress[32]. Starvation-dependent expression of the stressosome gene cluster and the presumed output module (*Vv*D1-D2) in minimal media was confirmed by analysis of the *VvrsbR*, *VvrsbT*, *VvrsbX*, *VvD1* and *VvD2* transcript levels by Northern-blotting (Supplementary Fig. 2a–c). As the Northern blot experiments show a lower transcription of the stressosome operon in the low iron condition, a possible link between iron-availability and stressosome expression requires further investigation. Taken together, these results show that when *V. vulnificus* grows in a minimal medium the stressosome is expressed upon exhaustion of nutrient and energy sources.

### The *Vv*RsbR and *Vv*RsbS proteins form the core of the minimal stressosome.
The differences between the *rsbRST*-module of *B. subtilis* and the *rsbRSTX*-module of *V. vulnificus* prompted a detailed analysis of the underlying direct interactions of the proteins involved. To do so, Bacterial-Two-Hybrid (BACTH) assays[33] were performed with the stressosome proteins *Vv*RsbR, *Vv*RsbS, *Vv*RsbT and the predicted stressosome phosphatase *Vv*RsbX. The BACTH assay supported interactions between *Vv*RsbR and *Vv*RsbS, *Vv*RsbS and *Vv*RsbX as well as self-interactions for *Vv*RsbR and *Vv*RsbX (Supplementary Fig. 3a, b). Only weak self-interaction for *Vv*RsbS was observed and reproducible interactions of *Vv*RsbT with *Vv*RsbS or *Vv*RsbR individually were not detected. Lack of VvRsbT binding is in accord with results from the *B. subtilis* stressosome, where the binding of RsbT was dependent on the formation of the RsbR:RsbS complex[3], a condition not captured by the BACTH assay. In addition, BACTH analyses were carried out in aerobic conditions, which might affect the binding of VvRsbT to VvRsbR.

To investigate if in the absence of any obvious RsbR paralogs from the *V. vulnificus* genome sequence, other STAS-domain proteins (VV1_0681, VV1_2658, VV2_1159, VV2_1170) similar in size to RsbS might contribute to stressosome formation, we probed their binding to *Vv*RsbR, *Vv*RsbS and *Vv*RsbT. The BACTH assay did not detect interactions between *Vv*RsbR, *Vv*RsbS or *Vv*RsbT with any of the other four small STAS domain proteins identified in the *V. vulnificus* genome, suggesting that these proteins do not play a role in stressosome formation in this species (Supplementary Figs. 3c–e and 4), and providing support for the specificity of the interactions observed by BACTH.

To investigate whether *Vv*RsbR and *Vv*RsbS interact in vivo, immuno-precipitation experiments were performed with protein

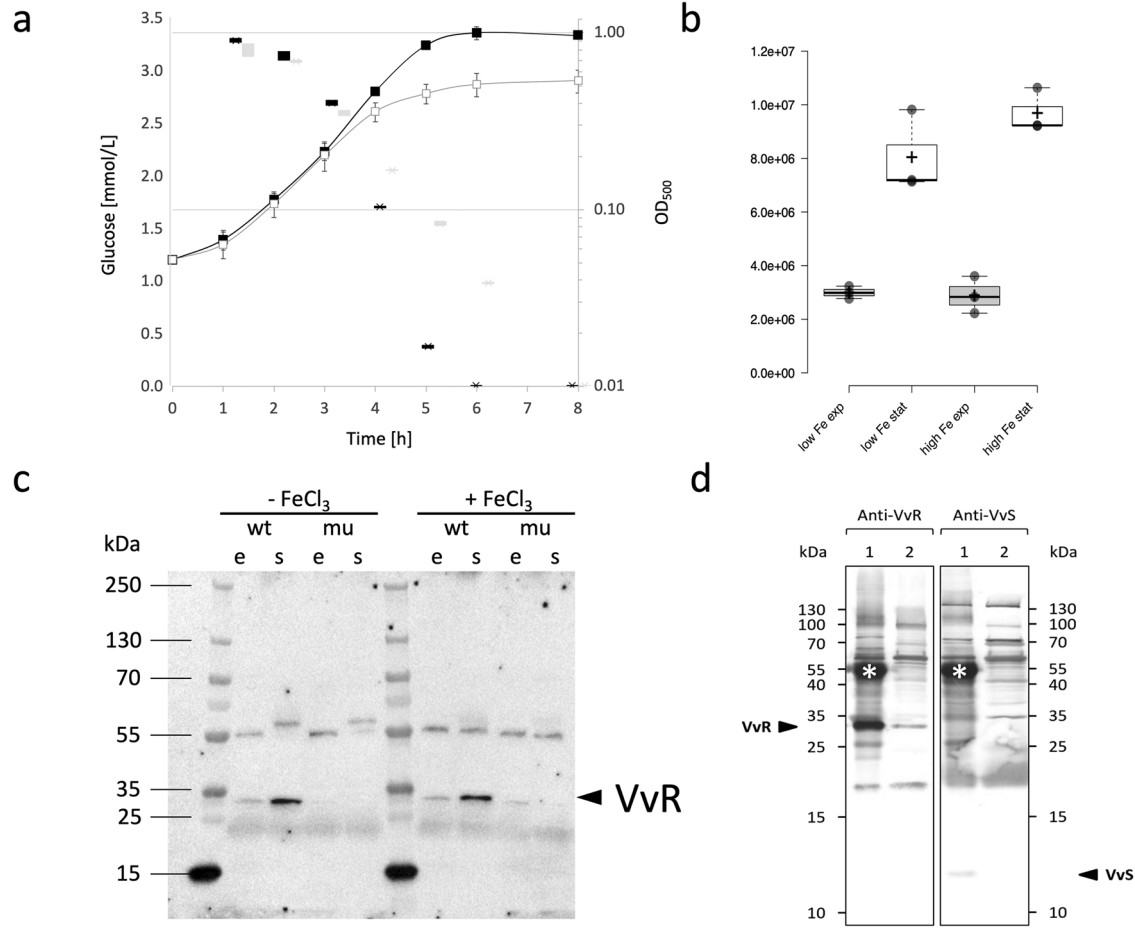

**Fig. 1 Stationary phase induction of the *V. vulnificus* stressosome. a** *V. vulnificus* was grown in minimal medium with (black symbols) and without (gray symbols) iron supplementation and the cell culture medium was analyzed for glucose consumption: glucose concentration (box plots) and optical density (lines with squares, with error bars showing standard deviations of three independent experiments). In addition, samples for Western blot analysis of *Vv*RsbR were taken during exponential growth and four hours after entry into stationary phase. Quantified *Vv*RsbR signals are shown in (**b**) and a representative uncropped Western blot is shown in (**c**); (e = exponential sample, s = stationary phase sample, wt = *V. vulnificus*) CMCP6 wild type and mu = isogenic ΔRSTX mutant. Box-whisker in (**a**) and (**b**) extend to data points that are less than 1.5 x interquartile ranges away from 1st/3rd quartile (Tukey). **d** For the detection of *Vv*RsbR:*Vv*RsbS complexes, the soluble protein fraction from stationary phase cells grown in iron supplemented minimal medium was subjected to immune precipitation using *Vv*RsbR (*Vv*R) specific polyclonal serum. Precipitated complexes (lane 1) and supernatant from the same precipitation reaction (lane 2) were separated by 1D SDS PAGE and blotted onto membranes. Membranes were then incubated with either anti-*Vv*RsbR (left two lanes) or anti-*Vv*RsbS (*Vv*S) (right two lanes) as primary antibodies. Bound *Vv*RsbR and *Vv*RsbS specific antibodies (arrows) were detected with anti-rabbit IgG. In addition to anti-*Vv*RsbR and anti-*Vv*RsbS the secondary anti-rabbit IgG also detected a band around 55 kDa (asterisk) in the precipitation reaction, which corresponds to the heavy chain of the anti-*Vv*RsbR antibody used for precipitation. Source data for (**a**) and (**b**) are available as supplementary data 1 and the uncropped blot from the immune precipitation experiment (**d**) is shown in Supplementary Fig. 18a.

extracts of *V. vulnificus* cultured in minimal medium in the presence of iron. The clarified *V. vulnificus* cell lysate harvested at early stationary phase was precipitated with protein A-coated magnetic beads pre-incubated overnight with a *Vv*RsbR specific antibody. SDS-PAGE and Western blot analysis of the bound protein-antibody complex with anti-*Vv*RsbR and anti-*Vv*RsbS specific polyclonal sera revealed the simultaneous enrichment of both proteins. The majority of *Vv*RsbR was clearly detected in the immunoprecipitate (Fig. 1d). Analysis with the *Vv*RsbS specific antibody detected a band in the *Vv*RsbR precipitate but not the corresponding supernatant. Thus, *Vv*RsbR and *Vv*RsbS interact in vivo, supporting the hypothesis that stressosome complexes assemble in *V. vulnificus*.

**A heme-group in the N-terminal domain of *Vv*RsbR is assembled in *Vv*RsbR:*Vv*RsbS stressosomes.** In silico sequence analysis predicted that the N-terminal domain of *Vv*RsbR

encodes a heme-dependent globin coupled sensor domain[34]. To validate this prediction, the *Vv*RsbR:*Vv*RsbS complex was produced recombinantly in *E. coli* and the *Vv*RsbR and *Vv*RsbS proteins co-purified as stable stressosome complexes by size-exclusion chromatography (Fig. 2a). The purified *Vv*RsbR:*Vv*RsbS complex showed an intense red color, consistent with the presence of a coordinated heme group (Fig. 2a). Subsequently, a *Vv*RsbR construct of the N-terminal 165 amino acids was produced for recombinant expression in *E. coli*. The purified domain retained the red coloration characteristic of heme binding (Fig. 2b). Under oxidizing conditions, the UV/visible absorbance spectrum of the *Vv*RsbR:*Vv*RsbS complex displayed three distinct peaks typical for a Fe(II)-$O_2$ heme-dependent globin coupled sensor domain: the Soret peak (414 nm), the α-peak (543 nm) and the β-peak (578 nm)[34]. After deoxygenation with sodium dithionite, the maximum of the Soret peak shifted to 431 nm and the α and β peaks merged into a single peak with a maximum at 556 nm, which is consistent with the Fe(II) form (Fig. 2c). This

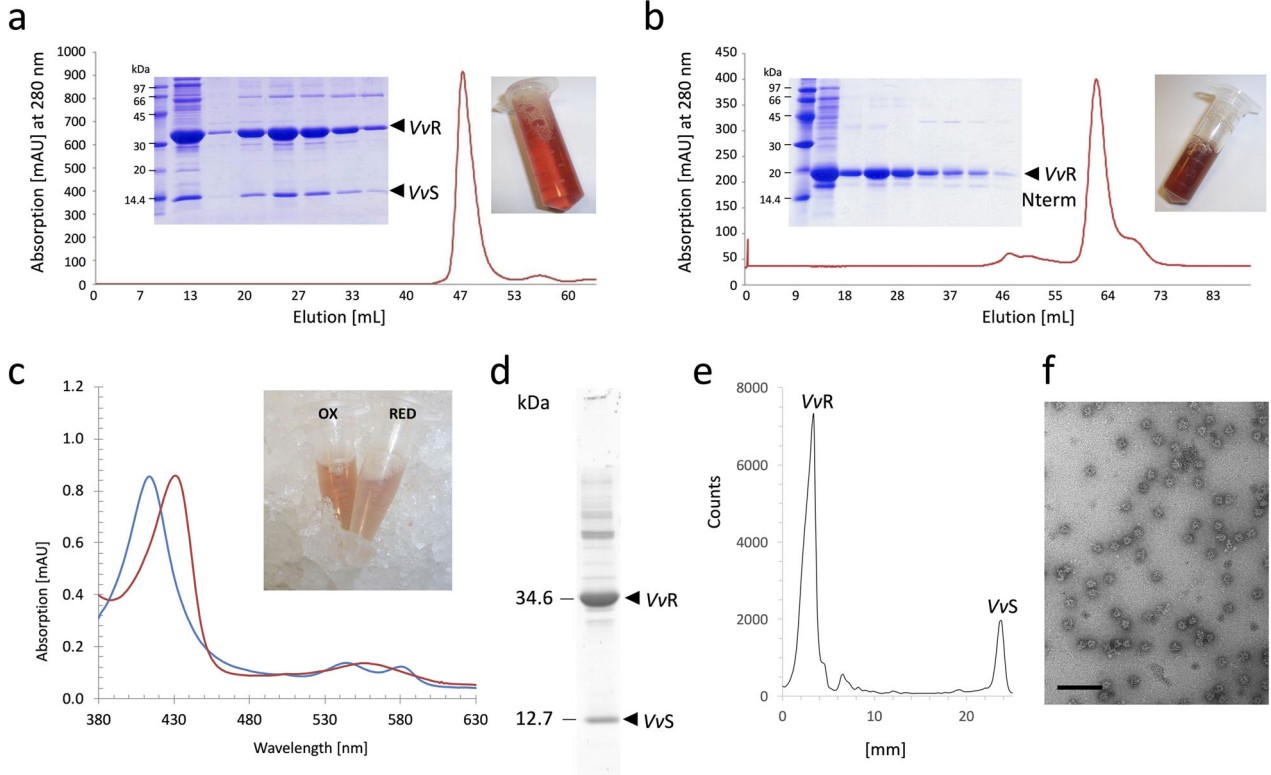

**Fig. 2 Biochemical characterization of the *Vv*RsbR:*Vv*RsbS complex.** Size exclusion chromatography profiles of the *Vv*RsbR:*Vv*RsbS complex (**a**) and the *Vv*RsbR N-terminal domain (**b**). Insets show Coomassie-stained fractions of the elution maxima after separation by SDS PAGE and the purified protein after concentration: *Vv*RsbR:*Vv*RsbS complex (13 mg/mL) and VvRsbRN-terminal domain (amino acid residues 1–165, 15 mg/mL). **c** Absorption spectra of oxygenated (blue line) and deoxygenated (red line) forms of the *Vv*RsbR:*Vv*RsbS complex. Deoxygenated *Vv*RsbR:*Vv*RsbS samples were prepared by adding sodium dithionite to the protein solutions. Spectra were obtained using a 1 cm path length quartz cuvette and a Biochrom Libra S22 UV-Vis spectrophotometer. The inset shows photographs of protein solutions in oxygenated (OX) and reduced (RED) states. **d** Purified *Vv*RsbR:*Vv*RsbS stressosome complex separated by SDS PAGE (uncropped gel lane) and stained with Krypton (Thermo Scientific). The gel was imaged with a Typhoon fluorescence scanner, with excitation at 532 nm and emission recorded at 560 nm. Band intensities were quantified with ImageQuant (**e**). The calculated ratio between *Vv*RsbR and *Vv*RsbS was 2.4:1. The uncropped gel is shown in Supplementary Fig. 18b. Electron micrographs of negative stained purified *Vv*RsbR:*Vv*RsbS complexes showed homogenously distributed particles (**f**). The size bar represents 100 nm.

shift in the spectrum was accompanied by a change in color of the protein solution from red (oxidized) to yellow (reduced; Fig. 2c). Therefore, the *Vv*RsbR:*Vv*RsbS stressosome complex binds an iron-heme cofactor through the N-terminal sensory domain of *Vv*RsbR, consistent with heme binding of *V. brasiliensis Vb*RsbR[30].

**The *V. vulnificus* stressosome is linked to iron metabolism.** To investigate the physiological function of the stressosome gene cluster, we investigated the proteome of a Δ*rsbRSTX* mutant in comparison to its isogenic wild type strain. First, we analyzed the proteome of exponentially growing and stationary cells cultured in iron-supplemented medium, a condition leading to strong accumulation of *Vv*RsbR in the stationary phase. Second, to study the predicted association with oxygen signaling, we characterized the proteome of exponentially growing cells transferred to screw top tubes, where hypoxic conditions are established by oxygen depletion of the growing cells (Supplementary Fig. 5a). These comparisons identified a large number of proteins with significantly changed abundance in response to growth phase and oxygen-availability, and confirmed the stationary phase accumulation of *Vv*RsbR and the two potential downstream signaling proteins, *Vv*D1 and *Vv*D2 (Supplementary Fig. 5b). Significantly, the abundance of several proteins differed between the wild type and the Δ*rsbRSTX* mutant in each condition (Supplementary

Fig. 5c, Supplementary data 2). The number of proteins with significantly changed abundance (1.5-fold and $p < 0.05$) in the Δ*rsbRSTX* versus wild type comparison was much smaller in exponentially growing cells than in stationary phase cells (Supplementary Fig. 5c). This suggests that a stationary phase signal, correlating with exhaustion of glucose, triggers a strong stressosome response. Following the hypoxic-shift, the lack of the stressosome caused a more pronounced down-regulation than upregulation of proteins in the stressosome mutant at all time points (Supplementary Fig. 5c), pointing to the importance of the stressosome to maintain protein expression in oxygen-restricted conditions. Although in all three conditions—growth, stationary phase and hypoxia—the loss of the stressosome resulted in pronounced changes in protein expression, little overlap in the stressosome-dependent proteomic signatures was observed (Supplementary Fig. 5d, Supplementary data 3 and 4).

Classification of proteins whose abundance was affected by the *Vv rsbRSTX* deletion revealed a functionally diverse set of targets covering various cellular processes e.g. metabolism, biosynthesis of secondary metabolites, cofactors and siderophores, ABC-transport, and signaling (supplementary data 2).

Apart from a great functional diversity of regulated proteins, links between the *Vv rsbRSTX* gene cluster and iron/heavy metal metabolism were observed. Examples of differentially abundant iron-metabolism proteins include the iron-uptake proteins FeoAB

**Fig. 3 Phylogenetic distribution of RSTX-modules with a sensor globin coupled RsbR ortholog.** The first row shows the domain organization of proteins encoded in the *B. subtilis* SigB operon. Here, the RST-module is followed by a gene encoding the environmental stress phosphatase RsbU. RsbU is activated upon interaction via its N-terminal domain (RsbU_N) with RsbT. Activated RsbU then conveys the signal to the RsbV-RsbW-SigB partner-switching module. RsbX, the last protein encoded in the operon, functions as a feedback phosphatase, reducing stressosome activity by the dephosphorylation of conserved threonine and serine residues in RsbR and RsbS. In contrast to *B. subtilis*, in species encoding a sensor globin coupled R-subunit, the putative feedback phosphatase, RsbX, is encoded immediately downstream of the RST-module thus forming a highly conserved RSTX-module. Based on the domain organization, eleven putative downstream architecture types were identified for these species. Some, but not all, of the downstream-located signaling proteins possess a RsbU_N like domain, supporting the notion that they may indeed function as targets for the RsbT-like protein. Domain abbreviations are as follows: RsbR_N (RsbR non-heme globin N-terminal domain), RsbU_N (RsbU N-terminal domain), PAS (Per- period circadian, Arnt- Ah receptor nuclear translocator, Sim- single-minded protein), GGDEF (Diguanylate cyclase, synthesizes cyclic di-GMP), HDc (Metal dependent phosphohydrolases with conserved 'HD' motif), STAS (Sulfate transporter and anti-sigma factor antagonist), PP2C ($Mg^{2+}$ or $Mn^{2+}$ dependent protein phosphatase 2C), HisKA (His kinase A (phosphoacceptor) domain), PAC (PAS associated C-terminal), EAL (Putative diguanylate phosphodiesterase, called EAL after its conserved residues), HATPase_c (Histidine kinase-like ATPase), Sigma (RNA polymerase sigma subunit), REC (CheY-homologous receiver domain), GAF (cGMP-specific phosphodiesterases, adenylyl cyclases and FhlA) and HPT (Histidine phosphotransfer domain).

(VV1_0148, VV1_0149), vulnibactin[35] siderophore synthesis and uptake proteins (VV2_0830, VV2_0831, VV2_0834, VV2_0835, VV2_0837, VV2_0838, VV2_0842, VV2_0843), ferric iron uptake ABC transporters (VV1_1660, VV1_1663), a TonB iron uptake system (VV2_0363, VV2_0364), a ferritin (VV1_1116), putative heme-iron utilization proteins (VV2_1616, VV2_1617), and proteins with a role in heavy metal resistance (VV2_0850, VV2_0853) (Supplementary Figs. 6–13). Expression of these proteins, except VV2_0850 and VV2_0853 which were upregulated, was lower in ΔrsbRSTX, suggesting a specific physiological impact of iron in the stressosome mutant. The link to iron metabolism was particularly evident during stationary phase, while only moderate further downregulation of iron related proteins in the ΔrsbRSTX mutant was observed in the hypoxia experiment.

Moreover, a consistent higher upregulation in the stressosome mutant was observed for an uncharacterized ABC-F protein (VV1_0491, between 25 and 29-fold) showing homology to the translation throttle Etta[36]. Additional proteins with a role in translation were differentially regulated in the mutant compared to the wild type. In stationary phase cells of the stressosome mutant we observed upregulation of the ribosome hibernation protein (VV1_0693, 4.7-fold), a peptidyl-tRNA hydrolase (VV1_0258, 4.3-fold) and proteins involved in tRNA biogenesis

and modification (VV1_0266, 2.2-fold; VV1_0277,2.1-fold). Finally, a potential translation release factor methyltransferase (VV1_0252, 2.9-fold) and several proteins with a role in tRNA modification (VV1_1251, VV1_2142, VV1_2608; VV1_2926) were present at lower level (1.5 to 1.9-fold) in the mutant, pointing to a potential impact of stressosome activity in modifying translation in *V. vulnificus* (supplementary data 2).

**Occurrence and genetic organization of globin coupled sensor containing stressosome gene clusters.** Iron-heme binding is a central feature of the stressosome in *V. vulnificus*, and to investigate whether heme-binding globin coupled sensors are a common feature of stressosomes, microbial genomic sequences were analyzed using *V. vulnificus* VvRsbR as the query (Fig. 3). The search identified many bacterial species possessing a *rsbRSTX*-module with an RsbR-encoded N-terminal globin coupled sensor. These species, ranging from Proteobacteria, Lentisphaerae, Bacteroidetes and Cyanobacteria, reside in aquatic ecosystems, in different types of marine or freshwater habitats. The single domain PP2C type phosphatase encoded in these gene clusters was closely related to RsbX but not RsbU[22]. Thus, these organisms all encode the minimal set of orthologs required to form a functional stressosome: an antagonist RsbS, a co-antagonist RsbR

with sensory function, the switch kinase RsbT, and the feedback phosphatase RsbX. Furthermore, several different types of potential down-stream modules were associated with the *rsbRSTX*-modules, which were usually separated only by a small number of base pairs (Fig. 3). The down-stream module domain organization suggests that in these organisms the stressosome has been adopted to control either: the activity of PP2C-type phosphatases (e.g. *Methylomicrobium*); the level of the secondary-messenger cyclic-di-GMP (e.g. *Vibrio*, *Chromobacterium*, *Herbaspirillum*); or phosphorelays of kinase receiver (HisKA) and transmitter (REC) domain proteins (e.g. *Dogia*, *Ruegeria*).

In some Proteobacteria of the α, β and γ group, the N-terminus of signaling proteins encoded downstream of the RsbX homolog shows significant sequence similarity to the N-terminus of the *B. subtilis* RsbU protein (RsbU_N). Since binding of RsbT to the RsbU_N transmits input from the stressosome to the downstream segment of the cascade regulating SigB in *B. subtilis*[18,19], this observation strongly suggests that these proteins contribute to stressosome dependent signaling. However, for the majority of the down-stream encoded signaling proteins considered in the analysis, including those identified in *V. vulnificus*, no RsbU_N domain was detected. Signaling by the stressosome in these cases could be transmitted through RsbT-dependent phosphorylation of the downstream sensor kinase, which would most likely be a transient interaction as required by a regulatory switch. This explanation is consistent with our failure to produce positive interaction results for VvRsbT and VvD1, the two-component protein encoded directly down-stream of the stressosome module. A screen of complete proteomes available at UniProt for the genus *Vibrio* identified the complete set of stressosome proteins (RsbR, RsbS, RsbT and RsbX) in a surprisingly small number of *Vibrio* species (four out of 38; supplementary data 5): *V. mangrove*, *V. nigripulchritudo*, *V. pectenicida* and *V. vulnificus*. These species all share a similar set of potential stressosome downstream regulators. Why stressosomes are restricted to only a few *Vibrio* species, and if this is a common phenomenon in other genera, is presently unknown.

**Single particle cryo-EM reconstruction of the *V. vulnificus* VvRsbR:VvRsbS stressosome reveals a new stressosome symmetry and stoichiometry**. Negative-stain transmission electron microscopy was performed on the purified VvRsbR:VvRsbS complex (Fig. 2d, e) to assess the formation of stressosome-like structures. The VvRsbR:VvRsbS complex was easily recognizable on micrographs due to its spherical core with several turret-like extensions, highly reminiscent of stressosome complexes from *B. subtilis* and *L. monocytogenes*[5,6,21] (Fig. 2f). Although oxidized and reduced stressosome complexes could be visualized in negative stain, only the oxidized form presented here was suitable for a cryo-EM study.

Quantification of VvRsbR and VvRsbS bands of the heterologously expressed stressosome complex revealed a non-integral VvRsbR:VvRsbS ratio of 2.4:1, suggesting a stoichiometry between VvRsbR and VvRsbS subunits that differs from the 2:1 RsbR:RsbS ratio reported for in vitro assembled *B. subtilis and L. monocytogenes*[5,21] stressosome complexes (Fig. 2d, e). To generate a high resolution structure of the *V. vulnificus* stressosome, and to identify the symmetry and stoichiometry of the VvRsbR:VvRsbS stressosome complex, a cryo-EM reconstruction was initiated from the oxidized stressosome complex (Table 1). Iterative 2D classifications of around 230,000 particles resulted in class-averages displaying a variety of symmetric features in the STAS domain core (supplementary fig. 14). Eigenimage analysis of the 2D classes showed patterns characteristic of icosahedral symmetry: twofold, threefold and fivefold

| | VvRsbRS complex (EMDB-12676) (PDB 7O01) |
|---|---|
| **Table 1 Cryo-EM data collection, refinement and validation statistics.** | |
| Data collection and processing | |
| Magnification | 59000 |
| Voltage (kV) | 300 |
| Electron exposure (e–/Å²) | 72 |
| Defocus range (μm) | −1.7 to −5.5 |
| Pixel size (Å) | 1.77 |
| Symmetry imposed | D2 |
| Initial particle images (no.) | 230,784 |
| Final particle images (no.) | 35,647 |
| Map resolution (Å) | 8.3 |
| FSC threshold | 0.148 |
| Map resolution range (Å) | 6.8–13.0 |
| Refinement | |
| Initial model used (PDB code) | 2MWG |
| Model resolution (Å) | |
| FSC threshold | 0.143 |
| Model resolution range (Å) | 8.3 |
| Map sharpening *B* factor (Å²) | |
| Model composition | |
| Non-hydrogen atoms | 36384 |
| Protein residues | 7356 |
| Ligands | — |
| *B* factors (Å²) | |
| Protein | mean 269.99 |
| Ligand | none |
| R.m.s. deviations | |
| Bond lengths (Å) | 0.015 |
| Bond angles (°) | 2.573 |
| Validation | |
| MolProbity score | 2.44 |
| Clashscore | 22.66 |
| Poor rotamers (%) | 0.0 |
| Ramachandran plot | |
| Favored (%) | 88.16 |
| Allowed (%) | 11.01 |
| Disallowed (%) | 0.83 |

symmetry axes (supplementary fig. 14). These features are especially prominent for the STAS domain core, and less pronounced when analyzing the whole complex. However, a comparison of the experimental 2D class averages with projections of an icosahedral-symmetric reconstruction clearly ruled out icosahedral symmetry for the VvRsbR:VvRsbS stressosome complex (supplementary fig. 14). Simultaneous 3D classification against multiple references with different combinations of two-, three- and five-fold symmetry axes (suggested by the Eigenimage analysis) resulted in a vast majority of particles being sorted into asymmetric (C1) or twofold symmetric (C2, D2) classes. Consequently, the following 3D reconstructions were performed either without applying symmetry (C1), or imposing D2 symmetry, which is supported by the observation of a D2 point-group symmetric arrangement of RsbR in *B. subtilis* stressosomes[5].

In contrast to the organization of the *Bacillus* stressosome, asymmetric reconstructions of the *Vibrio* stressosome show at least two pentagons consisting solely of VvRsbR monomers (Fig. 4), together with pentagons with 4:1 and also 3:2 VvRsbR:VvRsbS stoichiometry, respectively. Thus, the total number of VvRsbR₂ dimers in the complex can exceed 20 (Fig. 4), which is in agreement with the experimentally determined, non-integer ratio of 2.4:1 VvRsbR:VvRsbS. C1 refinements of subpopulations of the particles yielded reconstructions with varying

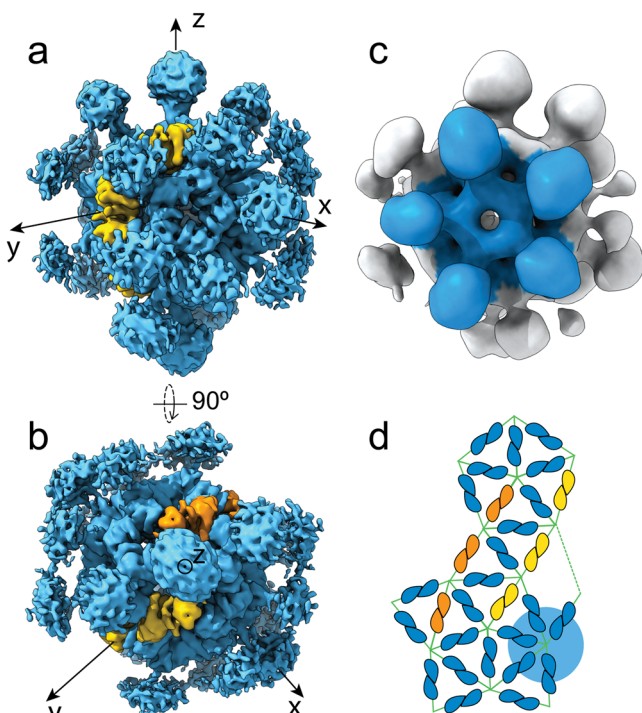

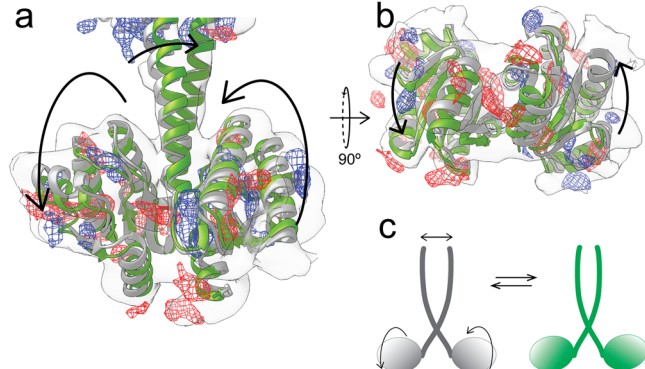

**Fig. 4 Cryo-EM reconstruction of the *Vibrio vulnificus* stressosome complex (*Vv*RsbR:*Vv*RsbS) reconstructed in D2 (a, b) and in C1 (c). a** The *Vv*RsbR subunits are colored in blue and *Vv*RsbS subunits are shown in yellow and orange. **b** The reconstructed volume reveals an imperfect D2 symmetry. **c** In the volume reconstructed without applying any symmetry operators, pentameric *Vv*RsbR faces are observed (colored blue in c and highlighted by the blue sphere in (**d**)). **d** The 2D representation of the icosahedron pentagonal faces reveal that only 12 hetero-triangular faces are formed in *Vv*RsbR:*Vv*RsbS, while 8 triangular faces are composed entirely of *Vv*RsbR.

**Fig. 5 Conformational changes of the *Vv*RsbR dimer protomers within the *Vv*RsbR:*Vv*RsbS stressosome.** Side (**a**) and bottom (**b**) views of a difference map between different *off-axis Vv*RsbR dimers (blue mesh indicates negative difference density, and red mesh indicates positive difference density). The difference map is superposed on an isolated dimer density shown as transparent surface. The green and white ribbon models indicate two putative conformations of the dimer. The schematic (**c**) indicates the changes in the respective positions of the monomers imposed by the transition.

numbers of turrets with an average of $22 \pm 3$ *Vv*RsbR$_2$ dimers, again clearly pointing towards a variability in *Vv*RsbR:*Vv*RsbS stoichiometry. Consistently, a stoichiometry of 22 RsbR$_2$ dimers was also reported for a recent 4.1 Å recombinant *B. subtilis* stressosome reconstruction[6].

*D2 symmetry break in the VvRsbR:VvRsbS complex* The cryo-EM single particle analysis in C1 symmetry revealed a 24:6 *Vv*RsbR$_2$:*Vv*RsbS$_2$ stoichiometry for the stressosome (Supplementary Fig. 15a). However, when applying D2 symmetry, we observed different reconstruction results depending on where the D2 axis was positioned. Changing the orientation of the D2 axis relating two triangular faces, with respect to the orientation of the x- and y-axes, resulted in changes in the stoichiometry of *Vv*RsbR and *Vv*RsbS, ranging from 20:10 *Vv*RsbR$_2$:*Vv*RsbS$_2$ to 18:12 *Vv*RsbR$_2$:*Vv*RsbS$_2$ (Supplementary Fig. 15b, c). In reconstructions with 20-18 *Vv*RsbR$_2$ dimers, the structural features in the STAS domain core were less prominent and it was not entirely possible to distinguish between *Vv*RsbR and *Vv*RsbS due to a potential misalignment. As expected, the *Vv*RsbR$_2$ dimers located on the D2 axis show symmetric features, while off-axis *Vv*RsbR dimers in all D2 maps exhibited an asymmetric sensory domain. Indeed, a difference map between the two protomers within an asymmetric *Vv*RsbR dimer reveals different states in the sensing domain, the coiled-coil linker helices and the STAS domain (Fig. 5), displaying a 5 Å 'shrugging' movement. Consequently, both protomers of the *Vv*RsbR are proposed to move either concertedly, in a scissor-like movement, or only one of the

protomers moves by skewing one of the linker's coiled-coil helices in the linker.

**Interactions within the *Vv*RsbR:*Vv*RsbS:*Vv*RsbT complex.** To complement the experimental data on *Vv*RsbR and *Vv*RsbS interactions, and to visualize the interaction with *Vv*RsbT, molecular models of *Vv*RsbR$_2$ and *Vv*RsbS$_2$ dimers and the *Vv*RsbT monomer were built. As stressosome particles were purified under aerobic conditions, and in agreement with the UV/visible absorption spectroscopy data, the sample is oxygen-bound, representing the inactive RsbT binding state of the stressosome[30]. The resolution of the refined D2 map extends below 7 Å in the core assembled from the STAS domains of the *Vv*RsbR and *Vv*RsbS and up to 8–9 Å in the distal N-terminal regions, showing well distinguished linker-helices in *Vv*RsbR and the beta-sheets in *Vv*RsbS in stressosomes with a 24:6 *Vv*RsbR$_2$:*Vv*RsbS$_2$ stoichiometry (supplementary fig. 16). Secondary structure elements in the STAS domain core and the protruding *Vv*RsbR linker helices are thus well defined and allowed for cryo-EM density guided model building of the *Vv*RsbR$_2$ and *Vv*RsbS$_2$ dimers (Fig. 6).

While the contact area between the monomers in *Vv*RsbS$_2$ comprises charged/polar residue chains, the same region in the *Vv*RsbR$_2$ dimer consists exclusively of hydrophobic side-chains (Fig. 6b, e). The charged/polar residues in *Vv*RsbS, e.g., interaction of Ser4-A and Ala6-A with Glu21-B, are likely to establish interactions between the monomers but are not conserved in *Vv*RsbR (Fig. 6c), resulting in a stronger interaction between the monomers of the *Vv*RsbS$_2$ dimer over the *Vv*RsbR$_2$ dimer. Notably, this interaction difference is very well conserved in RsbR and RsbS proteins from different Gram-negative bacteria but not conserved in Gram-positive bacteria (back-to-back submission Miksys *et al.*, in revision, COMMSBIO-21-1365A).

We generated a homology model of *Vv*RsbT and placed it onto the *Vv*RsbR$_2$:*Vv*RsbS$_2$ complex (Fig. 7) using the structure of the SpoIIAB:SpoIIAA complex[37] as a reference. The SpoIIAB serin kinase and its target SpoIIAA are functionally related structural homologs of the RsbT kinase and its RsbS substrate, forming a partner-switching module controlling activity of the sporulation sigma factor, $\sigma^F$, in Bacilli[37]. Although the ATP-loop in *Vv*RsbT is slightly shorter compared to related kinases, and consequently *Vv*RsbT is slightly more compact, up to 6 *Vv*RsbT molecules can

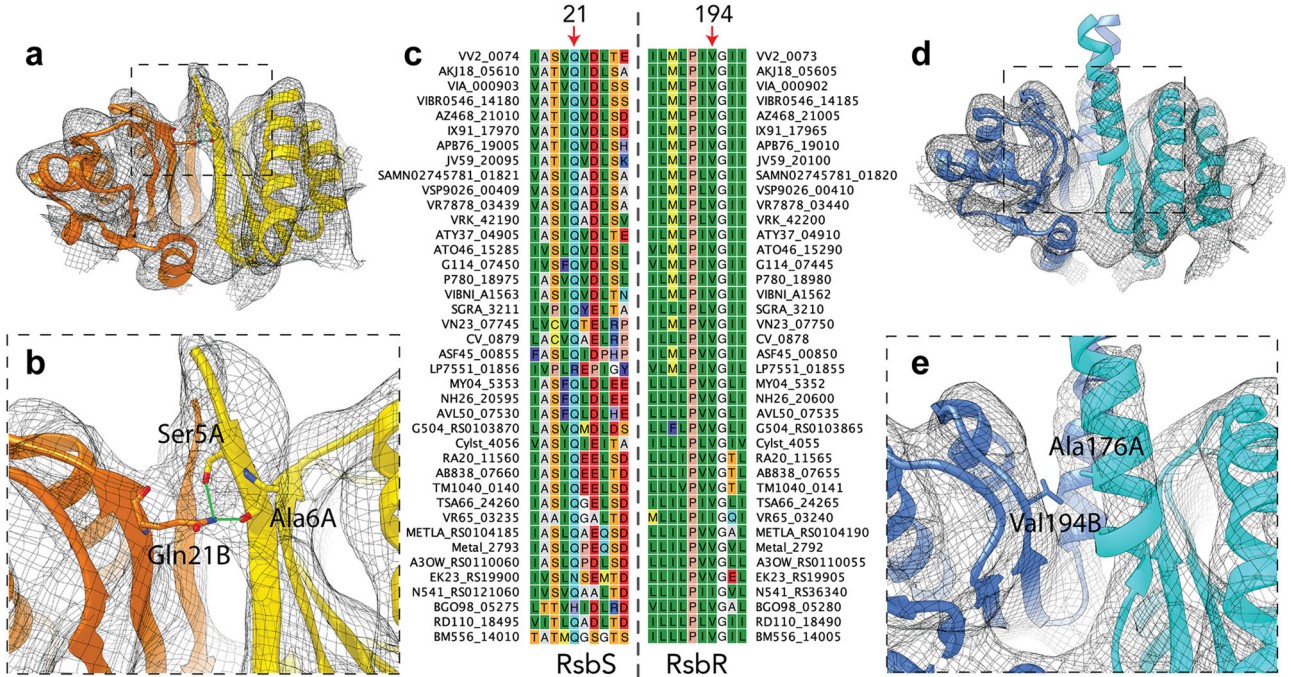

**Fig. 6 Interactions within the *Vv*RsbS and *Vv*RsbR STAS-domain interfaces and conservation of the involved residues within Gram-negative Bacteria.** **a**, **b** *Vv*RsbS and (**d**, **e**) *Vv*RsbR are shown in different shades of yellow and blue, respectively, fitted into the D2 cryo-EM density map (shown as black mesh). Protomers within the dimers of (**a**) *Vv*RsbS form intradimeric contacts via the β1-sheets (**b**) that likely result in increased stability of the dimer in comparison to *Vv*RsbR (for the residues contributing to the contacts, carbon atoms are colored blue or yellow/orange respectively, oxygen atoms in red, and nitrogen atoms in dark blue). **c** Gln21 in *Vv*RsbS is conserved in RsbS homologs from other Gram-negative bacteria. **d** Unlike *Vv*RsbS, *Vv*RsbR forms α-helices at the intradimeric interface that continue into the N-terminal turret-forming domain without obvious inter-monomeric contacts. **e** Val194 in *Vv*RsbR is conserved in other RsbR homologs (**c**) and does not feature a clustering of polar sidechains (hydrophobic residues are shown on a green background and polar charged, charged+, charged−, aromatic, are shown on light blue, dark blue, red and violet background, respectively).

be docked to the *Vv*RsbR₂:*Vv*RsbS₂ and *Vv*RsbS₂:*Vv*RsbR₂ interface in a stressosome (Fig. 7). There is not a pronounced complementarity of charges between the predicted *Vv*RsbT and *Vv*RsbR binding surfaces, consistent with the necessity of releasing *Vv*RsbT to elicit signal transduction (Supplementary Fig. 17a). In the RsbT bound pose of *Vv*RsbR only Thr233 (structural equivalent to *Bs*RsbR Thr205) would be easily accessible to the ATP-binding site in *Vv*RsbT, the second phosphorylation site in *Vv*RsbR, Ser199 (structural equivalent to *Bs*RsbR Thr175), is inaccessible (Fig. 7). Ser178 and Thr182, for which in vitro phosphorylation was reported for *V. brasiliensis* *Vb*RsbR[30] are buried within the linker-STAS domain interface in our *Vv*RsbR₂ model and would thus not be readily accessible to the *Vv*RsbT kinase in the assembled stressosome (Supplementary Fig. 17b).

## Discussion

Biochemical, functional and structural understanding of stressosome complexes is currently restricted to the low-GC Gram-positive *B. subtilis*[5,6] and *L. monocytogenes*[21]. The role of the stressosome in high-GC Gram-positives and Gram-negatives is largely unexplored. Recent pioneering work in *Mycobacterium marinum* showed that expression and cellular localization of stressosomes varies in response to stress and growth conditions in this species[38,39].

The *Vibrio basiliensis* stressosome complex is currently the only example studied from a Gram-negative bacterium. However, expression and phosphorylation of stressosome core protein *Xc*RsbR was also reported in the Gram-negative plant pathogen *Xanthomonas campestris* during transition from the late exponential growth phase to the stationary[30,40]. Here, we investigate the stressosome complex of the Gram-negative human

pathogen *Vibrio vulnificus*. We show that despite common features and principles shared with stressosomes from Gram-positives (*Bacillus/Listeria*-type), functionally relevant differences can be observed for the stressosome of Gram-negatives (*Vibrio*-type).

Significant differences between the *Bacillus/Listeria* and the *Vibrio* stressosomes are already evident at the genomic level[22]. The *V. vulnificus* genome demonstrates (i) an N-terminal globin coupled sensor of RsbR that is absent from *Bacillus/Listeria*, (ii) lack of a gene encoding RsbU downstream of the *rsbRST* module, (iii) stressosome locus association with genes related to two-component systems, instead of a sigma factor, and (iv) lack of *Vv*RsbR paralogs. Transcriptional analyses of the stressosome locus show that the two predicted downstream encoded signaling proteins, *Vv*D1 and *Vv*D2, follow the same transcription kinetics as the *rsbRSTX* module, supporting the idea that these genetic elements are functionally coupled. Furthermore, transcription of the stressosome locus increased in stationary phase after depletion of the main carbon source, contrasting results obtained for *B. subtilis*[32].

A global proteomics-based profiling of a wild type strain and an isogenic mutant lacking *Vv*rsbRSTX was used to identify the physiological function of the stressosome in *V. vulnificus*. Although our proteome analysis detected *Vv*RsbR during all growth conditions in the wild type, showing strongest accumulation during stationary phase in accordance with our Western blot results, there is currently an absence of a direct indicator/reporter for the degree of stressosome activation (e.g. changes in the stressosome phosphorylation pattern). We observed a clear difference in the proteome between wild type and the Δ*rsbRSTX* mutant, which is particularly strong when the stationary phase is triggered by glucose exhaustion in iron-rich medium. A

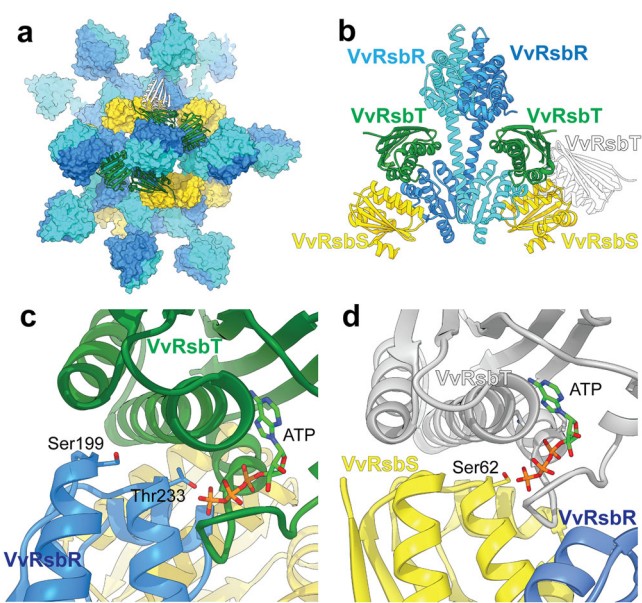

**Fig. 7 Interaction of *Vv*RsbT with phosphorylation sites in *Vv*RsbR and *Vv*RsbS.** A homology model of *Vv*RsbT was generated and docked to *Vv*RsbR and *Vv*RsbRS. **a** A space-filling model of the *Vv*RsbRS stressosome complex is shown with *Vv*RsbR₂ and *Vv*RsbS₂ dimers colored in blue and yellow, respectively. *Vv*RsbT (ribbon model) was docked to the *Vv*RsbR₂ (*Vv*RsbT in green) or *Vv*RsbS₂ (*Vv*RsbT in white). **b** Side view of the *Vv*RsbR₂ dimer and two *Vv*RsbS monomers with docked *Vv*RsbT colored as in (**a**). Close-up views show that binding of *Vv*RsbT to neighboring *Vv*RsbR or *Vv*RsbS is possible without steric clashes. The ATP binding site in *Vv*RsbT docked to *Vv*RsbR (**c**) and *Vv*RsbS (**d**) in the stressosome assembly. The respective phosphorylation sites are indicated (carbon atoms in blue or yellow respectively, oxygen atoms in red, and nitrogen atoms in dark blue).

prominent group of proteins linked to iron-metabolism and -uptake showed a consistent decrease in abundance in the stressosome mutant in all growth conditions tested. This finding could point to a reduced requirement for iron in the absence of an active stressosome or in a reduced capacity to induce iron-uptake pathways. At the very least the data suggest that the stressosome alters iron-metabolism in *V. vulnificus;* which could be relevant given the stressosome gene cluster is often associated with clinical isolates of *V. vulnificus*, as high blood iron is a risk factor for the development of severe *V. vulnificus* infection[41]. The oxygen-depletion signature of the mutant showed little overlap with the stationary phase signature. These differences between the oxygen-depletion response and the response to energy limitation might indicate that the *V. vulnificus* stressosome signaling-pathway integrates further signals. Indeed, a recent study linked the aerotaxis globin-coupled sensor domain of HemAT to ethanol-sensing[42], and the presence of a PAS domain in the presumed output module protein VvD1 (VV2_0077) could provide a further entry point into the signaling pathway.

We have determined the cryoEM structure of the stressosome from a Gram-negative organism in three symmetry states, C1, C2 and D2, and compared and contrasted these structural data with prior structural analyses of stressosomes from Gram-positive microbes.

First, the *Bs*RsbR:*Bs*RsbS[5,6] and *Lm*RsbR:*Lm*RsbS[21] stressosome reconstructions share the maximal number of 20 hetero-triangular capsid faces, while *Vv*RsbR:*Vv*RsbS in the most common symmetry, D2, comprises only 12 hetero-triangular faces. Because the *Vv*RsbR turrets prevent access to the STAS core, the 8 *Vv*RsbR homo-triangular faces cannot contribute to *Vv*RsbT

binding. Moreover, due to significant differences at the inter-dimeric interaction sites in *Vv*RsbR₂:*Vv*RsbS₂, it is not possible to fit a second *Vv*RsbT to a *Vv*RsbR at the *Vv*RsbS contact site, as is possible in the *Listeria innocua Li*RsbR:*Li*RsbS structure (Miksys *et al.*, in revision, COMMSBIO-21-1365A). Consequently, only one *Vv*RsbT molecule can be bound at the *Vv*RsbR:*Vv*RsbR interface, suggesting a greatly reduced capacity to interact with *Vv*RsbT. Any variation in the number of hetero-triangular capsid faces would provide an additional regulatory mechanism to titrate bound *Vv*RsbT, thereby modulating the downstream signaling cascade. It should be considered here, however, that the minimal VvRsbR:VvRsbS complex investigated in this study has been heterologously expressed and purified from *E. coli*, which is also the case for all other published stressosome structures. It is thus possible that the stoichiometry adopted by native stressosome complexes is different to these recombinant stressosomes.

Our structural data also provide insight into the mechanism of signal perception and transduction within the *Vv*RsbR₂ dimer. Considerable differences between the two monomers in the *Vv*RsbR₂ dimer were observed, presumably representing two different states of the *Vv*RsbR protein. This finding is reminiscent of observations previously reported for HemAT[43], an aerotaxis signal transducer protein that shares the heme-binding pocket with the N-terminal sensor domain of *Vv*RsbR. While in its ligand-bound form, the HemAT dimer assumes a highly symmetrical organization, the unoccupied form of the HemAT dimer displays a symmetry breach. This breach in symmetry is manifested in a small helical shift and rotation of the four-helical bundle in the dimerization interface, and more prominent in the ligand binding pocket of one of the two HemAT protomers. Disruption of the HemAT symmetry communicates the ligand binding state to the C-terminal domain of HemAT, and subsequently the downstream elements of the aerotaxis signaling cascade in *B. subtilis*. Asymmetry could be a specific mechanistic feature of oxygen sensing in globin coupled sensors and is reflected by the different states observed for *Vv*RsbR in our structure. Thus, as reported for HemAT, similar structural changes associated with ligand binding by *Vv*RsbR could be communicated by movements of the *Vv*RsbR linker helices and further transmitted to the STAS domain core, affecting the binding position and kinase activity of *Vv*RsbT. This could represent a fundamental aspect of stressosome-dependent signaling and should be considered in future analyses. Finally, we conclude that despite obvious differences in the molecular make-up of stressosome pathways in unrelated organisms, a recurrent theme in stressosome signaling is the global reorganization of the cell physiology to overcome adverse growth conditions.

## Methods

**Cultivation of bacteria**. *V. vulnificus* CMCP6[44] was grown in a chemically defined medium (7.5 mM glucose, 10 mM Na₂HPO₄, 10 mM KH₂PO₄, 0.8 mM MgSO₄, 9.3 mM NH₄Cl, 428 mM NaCl, 0.75 μM FeCl₃) at 30 °C with 120 rpm agitation. To stabilize the iron during storage, the FeCl₃ stock-solution was prepared in 1 N HCl. To neutralize the HCl transferred with the FeCl₃, an appropriate volume of 1 N NaOH was added to the final medium to archive final pH 7.0.

Exponentially growing cells from an overnight culture were used to inoculate fresh, pre-warmed medium with a starting OD₅₀₀ of 0.05. To analyze the impact of iron availability on stressosome expression, cells were also grown in medium without iron supplementation. To induced hypoxic conditions, an appropriate amount of exponentially growing cells was transferred to completely fill 50 ml Falcon tubes, leaving no residual air bubbles in the tubes. The filled Falcon tubes were subjected to static incubation at 30 °C and harvested at 30, 60 and 120 min after the shift.

**Bacterial two-hybrid screen**. For the bacterial two hybrid analyses[33] the coding sequences of *Vv*RsbR (VV2_0073), *Vv*RsbS (VV2_0074), *Vv*RsbT (VV2_0075), *Vv*RsbX (VV2_0076) and the predicted STAS domain proteins VV1_0681, VV1_2658, VV2_1159 and VV1_1170 were amplified by PCR from *V. vulnificus* CMCP6 genomic DNA and cloned into the XbaI and KpnI site of pUT18, pUT18C, pKT25 and p25-N to generate N- and C-terminal fusions with the T18

and T25 fragment of the *Bordetella pertussis* adenylate cyclase, CyaA. To test for protein-protein interactions, appropriate plasmid combinations encoding a T18 and a T25 fusion were co-transformed into *E. coli* BTH101 and transformants selected overnight at room temperature (RT) on LB plates supplemented with 50 μg ml$^{-1}$ kanamycin. The next day co-transformants were streaked on LB-agar plates containing 50 μg ml$^{-1}$ kanamycin, 100 μg ml$^{-1}$ ampicillin, 100 μg ml$^{-1}$ X-Gal (5-bromo-4-chloro-3-indolyl-β-D-galactopyranoside), and 0.5 mM IPTG (isopropyl β-D-1-thiogalactopyranoside). Plates were stored in the dark at RT and X-Gal degradation monitored for up to three days. pUT18Czip and pKT25-zip expressing a fusion of T18 and T25 with the leucine zipper of GCN4 were used as a positive control and the pUT18C and pKT25 empty vectors as a negative control. Colonies that turned blue following the above procedure in all three replicates were deemed as positive for protein-protein interactions due to the functional complementation of *cyaA*, which results in the expression of ß-galactosidase and the hydrolysis of X-Gal.

**RNA isolation and Northern blot analysis**. *V. vulnificus* was harvested by mixing 30 ml cell culture with 10 ml ice-cold killing buffer (20 mM Tris pH = 7.5, 5 mM MgCl$_2$, 20 mM sodium azide) and centrifugation at 21,000 $x$ g and 4 °C for 5 min. Isolation of RNA was performed as previously described using the acid-phenol method with modifications as described by Fuchs et al.[45] Briefly, the cells were resuspended in 0.5 ml ice-cold suspension buffer (3 mM EDTA pH 8.0, 200 mM NaCl), mixed with 0.5 ml μl PCI (phenol:chloroform:isoamyl alcohol, 25:24:1) and 0.5 μl glass beads (0.1 mm diameter) and lysed in a Precellys 24 (Peqlab, Erlangen, Germany) with one 30 sec cycle at 6800 rpm. After 5 min centrifugation at 21,000 $x$ g the aqueous phase containing the RNA was transferred to a new test tube and mixed with 500 PCI for 5 min in a shaker. The aqueous and the organic phase were separated by centrifugation for 5 min at 21.000 $x$ g and the upper, aqueous phase washed once with CI (chloroform:isoamyl alcohol, 24:1). After transfer of the RNA containing phase (usually 0.4 ml) to a new test tube the RNA was precipitated overnight at −20 °C with 1 ml ice-cold ethanol (98%) and 40 μl sodium acetate (3 M). The RNA was pelleted for 30 min at 21,000 $x$ g and 4 °C. The RNA pellet was washed with 0.5 ml ice-cold ethanol (70%) and dissolved at RT for 5 to 10 min in 50–200 μl distilled water to reach a final concentration of 1 to 2 μg per μl. RNA was stored at −70 °C.

For Northern blot analyses, total RNA was prepared from three independent experiments. Digoxygenin-labeled RNA probes were synthesized by in vitro transcription using T7 RNA polymerase and appropriate DNA fragments as templates. The DNA fragments were generated by PCR using appropriate primers (supplementary table 1) with chromosomal DNA of *V. vulnificus* CMCP6 as a template. Separation, transfer and detection of RNA were carried out as described previously[45].

*Western blot analysis* - Polyclonal antibodies against Strep-tagged *Vv*RsbR and *Vv*RsbS were raised in rabbits (Pineda, Berlin, Germany) and affinity- purified using CNBr-activated agarose coupled with the respective target protein.

*V. vulnificus* cells were re-suspended in 1 ml TE buffer (10 mM Tris/HCl, 1 mM EDTA, pH = 8.0) and disrupted with 0.5 ml glass beads of 0.1 mm diameter in a Precellys 24 (Peqlab, Erlangen, Germany) with two 30 sec cycles at 6800 rpm. To remove glass beads and cell debris the lysate was centrifuged for 15 min at 4 °C and 21,000 $x$ g. Afterwards the supernatant was subjected to a second centrifugation step at 4 °C for 10 min at 21,000 $x$ g. Protein extracts were stored at −20 °C.

Proteins were separated by 1D SDS-PAGE and transferred to PVDF membranes for 1.5 h at 250 mM and 150 V using standard procedures. *Vv*RsbR and *Vv*RsbS specific sera were used at a 1:5000 and 1:2000 dilution, respectively. Bound anti-*Vv*RsbR and anti-*Vv*RsbS antibodies were detected with a monoclonal alkaline phosphatase conjugated mouse anti-rabbit IgG at 1:200,000 dilution and with NBT and BCIP as substrates.

**Co-immunoprecipitation of *Vv*RsbR:*Vv*RsbS**. Stationary *V. vulnificus* cells grown in a minimal medium with iron supplement were harvested by centrifugation (10,000 × g, RT, 10 min) washed in TE buffer (1 mm EDTA, 10 mM Tris pH = 7.0) and lysed by vigorous agitation in the presence of glass beads (0.1–0.11 mm, Sartorius Stedium Biotech) by two cycles in a Precellys 24 device for 30 s at 6800 rpm (Bertin Technologies) in lysis buffer (1 mM EDTA, 10 mM Tris, 1.0% (v/v) Tween 20). Glass beads and cell debris were removed by two centrifugation steps at 21,000 × g at 4 °C for 10 min. The supernatant of the second centrifugation step was used for immuno-precipitation. 0.5 ml of protein solution was mixed with 10 μl *Vv*RsbR antibody (0.5 μg μl$^{-1}$) and incubated overnight in a rotary shaker at 4 °C. After overnight incubation, protein A coated magnetic beads (Dynabeads, Novex) from 50 μl bead solution were added to the protein antibody mix and incubated for additional 2 h at 4 °C. Beads were collected with help of a magnet and washed three times with 200 μl wash buffer (1x PBS (pH 7.4) and 1% (v/v) Tween 20). The supernatant was stored at −20 °C for further analysis. Antibodies and bound proteins were eluted from the beads by incubation with 20 μl elution buffer (50 mM glycine, pH = 2.8) and 10 μl 3x SDS-PAGE loading buffer for 10 min at 70 °C. Finally, beads were removed by magnetization and the supernatant loaded on a 15% SDS-PAGE for further analysis by Western blotting.

**Stressosome mutant construction**. In order to study the role of the *V. vulnificus* stressosome in vivo, a knock-out mutant lacking the *Vv*RsbR, *Vv*RsbS, *Vv*RsbT and

*Vv*RsbX genes was constructed (*V. vulnificus* Δ*RSBRSTX*). The synthetic knockout allele (Eurofins) had a deletion from nucleotide 4 of *Vv*RsbR to nucleotide 577 of *Vv*RsbX and was inserted into the pDS132 suicide vector[46] to create plasmid pDS_Δ*RSBRSTX*. The mutagenesis protocol was carried out by bi-parental conjugation of pDS_Δ*RSBRSTX* into *V. vulnificus* CMCP6 and then allowing two consecutive events of homologous recombination. The *E. coli* donor strain chosen for the bi-parental conjugation was β2163[47], auxotrophic for 2,6-Diaminopimelic acid (DAP). *V. vulnificus* CMCP6 and *E. coli* β2163 pDS_Δ$\overline{RSBRSTX}$ were grown overnight at 37 °C in LBN and LB + 25 μg/mL Chloramphenicol + 0.3 mM DAP, respectively. Cultures were washed in LB broth, recipient and donor cells were mixed at a 1:1 ratio (v/v) and spotted onto LB + 0.3 mM DAP agar plates and incubated for 5 h at 37 °C. Chloramphenicol-resistant *V. vulnificus* transconjugants carrying pDS_Δ*RSBRSTX* plasmid name integrated into the chromosome by homologous recombination were selected onto LBN + 5 μg/mL Chloramphenicol. The second event of homologous recombination was triggered by culturing the first recombinant cells in LBN broth without Chloramphenicol. Second recombinant cells were selected onto LBN + 5% sucrose (w/v). Due to the *sacB* gene present in the suicide plasmid, growth in the presence of sucrose can be achieved only after the second event of homologous recombination has efficiently excised the plasmid from the chromosome of *V. vulnificus*. This event can equally generate Wild-Type cells or deletion mutants. To select the stressosome mutant, colonies grown on LBN + 5% sucrose were screened by PCR using *Taq* polymerase (Bioline) and using the primers RSTX_For (5′-GTCACGGGTTGATTGATTCGCAT-3′) and RSTX_Rev (5′-CTCACCGAGACGTAACATATGAATGT-3′), mapping just outside the putative deletion site. The amplification was verified through agarose gel electrophoresis. Two different PCR products, of approximately 300 bp and 2600 bp, were expected for the Δ*RSBRSTX* mutant and WT, respectively. A Δ*RSBRSTX* mutant was selected and the mutation was confirmed through Whole Genome Sequencing (WGS) analysis (MicrobesNG).

**Cloning and expression of *V. vulnificus* stressosome proteins**. The full-length gene sequences of VvRsbR and VvRsbS were amplified by PCR from genomic DNA of *V. vulnificus* strain CMCP6. In a second PCR, the VvRsbR- and VvRsbS-encoding fragments were fused thereby introducing an additional ribosomal binding site (RBS) between the genes for VvRsbR and VvRsbS. The additional RBS was derived from that found naturally in front of the gene for VvRsbR and necessary to increase heterologous expression of VvRsbS in *E. coli*. The fused DNA fragment was cloned into the BsaI site of pPR-IBA1 (IBA, Göttingen, Germany) yielding pJPF012. For the co-expression of VvRsbR and VvRsbS as untagged proteins, a stop codon was introduced immediately downstream of the VvRsbS coding sequence. For expression and purification of the VvRsbR N-terminal domain the VvRsbR sequence was cloned into pBR-IBA1 using primers that introduced a stop codon within the VvRsbR sequence to yield a construct corresponding to the first 165 amino acids of VvRsbR. All primers are listed in Supplementary Table 1.

For over-expression of *V. vulnificus* stressosome proteins, the plasmid co-expressing VvRsbR and VvRsbS was transformed into *E. coli* TUNER (DE3) cells with ampicillin selection (100 μg ml$^{-1}$). Transformants were used to inoculate an overnight culture grown at 37 °C and 180 rpm in LB medium supplemented with ampicillin (100 μg ml$^{-1}$). To start the expression-culture, a two liter Erlenmeyer flask with one liter LB medium supplemented with ampicillin (100 μg ml$^{-1}$) was inoculated to a starting OD$_{600}$ of 0.05. Cells were grown at 37 °C and 180 rpm overnight without induction. Cells were harvested by centrifugation at 4 °C, for 30 min at 2,130 $x$ g and the cell pellet was stored at −80 °C.

For the production of *Vv*RsbR and *Vv*RsbS specific polyclonal sera, the respective full-length sequences were amplified by PCR with primers listed in supplementary table 1 from *V. vulnificus* genomic DNA and cloned as C-terminal Strep-tag fusions into the vector pPR-IBA1 (IBA, Göttingen, Germany) yielding vectors pEB02 and pEB13, respectively. Expression of *Vv*RsbR followed the protocol described above for the expression of the *Vv*Rsb:*Vv*RsbS complex. *Vv*RsbS expressing *E. coli* cells were induced with 1 mM IPTG when the culture reached an OD$_{540}$ of 0.6 and harvested 3 h after induction by centrifugation at 4 °C, for 30 min at 2.130 × g and the cell pellet was stored at −20 °C.

**Protein purification procedures**. The *Vv*RsbR:*Vv*RsbS complex was purified in a three-step procedure. Cells were resuspended in 30 ml low salt buffer (50 mM Tris/HCl, 150 mM NaCl, pH = 8.0) and disrupted by sonication with two cycles of 2 min with a Sonoplus sonicator (Bandelin, Berlin, Germany) with 50 % pulse and 80 % power. Cell debris and insoluble material were pelleted by centrifugation at 34,000 × g for 30 min. The cell free extract containing the soluble protein was mixed with solid ammonium sulfate to a final concentration of 15% and incubated for 30 min with stirring at 4 °C. After ammonium sulfate precipitation, the solution was cleared by centrifugation at 34,000 × g for 30 min at 4 °C. For further purification, the supernatant containing the *Vv*RsbR:*Vv*RsbS stressosome complex was filtered (0.45 μm) and loaded at a flow-rate of 3 ml min$^{-1}$ onto a Toyopearl Phenyl-650S hydrophobic interaction column (Tosoh, Tokyo, Japan) pre-equilibrated in buffer A (15 % (NH$_4$)$_2$SO$_4$, 50 mM Tris/HCl, pH = 8.0). Unbound protein was removed by washing with buffer A, before the column was developed with a 200 ml linear gradient of buffer B (50 mM Tris/HCl, pH = 8.0). Fractions containing the *Vv*RsbR:*Vv*RsbS complex were identified by SDS-PAGE and color

(because of the bound heme), pooled, and concentrated by ultrafiltration with a 30 kDa MWCO centrifugal concentrators (Millipore, Billerica, USA) at 4,000 x g and 4 °C before a final purification step by gel filtration chromatography using a HiPrep 16/60 Superdex S-200 column (GE Healthcare). The gel filtration column was pre-equilibrated with running buffer C (150 mM NaCl, 20 mM Tris HCl at pH = 8.0). Fractions of 2 ml were collected at a flow-rate of 1 ml min$^{-1}$. Fractions containing VvRsbR:VvRsbS were identified by color and stored at 4 °C. Purification of Strep-tagged VvRsbR and VvRsbS using Strep-tag columns was performed according to the protocol of the manufacturer (IBA, Göttingen, Germany).

**Measurement of VvRsbR:VvRsbS UV/visible absorption spectra**. Purified VvRsbR:VvRsbS complex was diluted to 1.4 µM final concentration in buffer C (150 mM NaCl, 20 mM Tris/HCl at pH = 8.0). Protein concentrations were determined using the Bradford assay. A standard curve using known quantities of bovine serum albumin was generated. The absorption spectra of air-oxidized and dithionite-reduced (0.8 mM final concentration) complexes were recorded with a Biochrom Libra S22 UV/Vis spectrophotometer (Biochrom Ltd.) in a 1 cm path length quartz cuvette in the 250–700 nm range. Measurement of the reduced complex was performed immediately after addition of the reducing agent and 20 minutes later, to re-evaluate the redox state of the sample.

**In silico analysis of V. vulnificus stressosome proteins**. To identify stressosome gene clusters encoding RsbR homologs with an N-terminal globin coupled sensor domain, the VvRsbR (VV2_0073) sequence of V. vulnificus CMCP6 was used as query in a protein-protein BLAST search (http://blast.ncbi.nlm.nih.gov/Blast.cgi) using default parameters. Hits were checked for the presence of an N-terminal globin coupled sensor domain (Protoglobin, Pfam: PF11563) and a C-terminal Sulfate Transporter and Anti-Sigma factor antagonist (STAS, Pfam: PF01740) domain using the SMART tool for the analysis of protein domain architectures[48]. Next, for protein hits with a Protoglobin-STAS domain architecture, genomic context analyses were manually carried out by inspection of the graphical representation of nucleotide data from the respective bacterial strain at NCBI (http://www.ncbi.nlm.nih.gov/nuccore) to identify complete RST-modules and in addition, to retrieve protein sequences of genes encoded up- and down-stream of the RST-module. Domain composition of all retrieved protein sequences was analyzed with SMART. Sequences were aligned with EXPRESSO[49] and displayed using ESPrit[50]. A manually curated set of bacterial RsbR, RsbS, RsbT and RsbX proteins was used to create a Hidden Markov model for each group and to screen all complete Vibrio proteomes provided by UniProt (release 2021_03) using HMMER (v3.1b2) with an e-value threshold of 1e-10[51,52]. Overlapping hits were resolved based on the model with the more significant e-value.

**Proteomic analysis - Cell disruption and protein lysate generation**. 16 OD units of cells were harvested by centrifugation, the supernatant was discarded and the resulting pellet was immediately frozen in liquid nitrogen and stored at −80 °C for further preparation. For the cell disruption the pellets were resuspended in 100 µL Tris-HCl 5 mM pH 7.4 containing 5% SDS each and immediately disrupted mechanically using in the Dismembrator/Retsch (ball mill) for 3 min at 2600 rpm (in a 4,8 ml Teflon vessel on liquid nitrogen with an 8 mm diameter steel ball). The cell powder was resuspended with 400 µl of preheated (95 °C) Tris-HCl buffer (5 mM pH 7.4) and the viscous lysate was transferred into a fresh 1.5 mL low bind pre-lubricated Eppendorf tube and shaken for 1 min at 95 °C and 1400 rpm. Then the lysate was cooled to room temperature and 2 µL of a 1 M MgCl$_2$ stock solution (final 4 mM MgCl$_2$) was added. Then 1 µL of a 1:100 diluted benzonase (Pierce Universal Nuclease No#88702) stock solution (final 0.005 U/µL) was added and mixed by short vortexing. The samples were then incubated at room temperature in an ultrasonic bath for 5 min until the viscous lysate was liquefied by complete degradation of DNA and RNA before the raw lysates were centrifuged for 30 min at 17000 g at room temperature. After centrifugation the protein lysate was transferred into a fresh 1.5 mL low bind pre-lubricated Eppendorf tube and the pelleted cell debris was discarded. Protein concentration of the samples was determined using the Micro BCA Protein Assay Kit following the manufacturer's protocol (Pierce, Rockford, IL, USA; prod. No. 23235) using a FLUOstar Omega Plate Reader (BMG Labtech). Samples were always stored at −80 °C. Sample preparation for mass spectrometry measurements was performed using the SP3 protocol as described in Blankenburg et al., 2019[53].

**LC-MS analysis**. The measurement of samples was performed on a LC-MS/MS platform containing reversed phase nano liquid chromatography (nano Acquity M-class UPLC, Waters corporation) coupled to nano spray ionization tandem mass spectrometry with traveling wave ion mobility using high-definition data independent (HD-MSE) acquisition and enabled with hybrid quadrupole orthogonal acceleration time of flight mass spectrometer (Synapt G2Si, Waters Corporation). The peptide mixture was separated on ACQUITY UPLC® M-Class HSS T3 1.8um, 75um x 200 mm column (Waters Corporation) using mixture of two buffers A and B (Buffer A, 0.1% (v/v) acetic acid in water; Buffer B, 0.1% (v/v) acetic acid in acetonitrile) by formation of a gradient with an increasing concentration of Buffer B at a flow rate of 300 nl/min from 5–26% (v/v) B in 170 min. The eluents sprayed at a voltage of 1.85–1.90 kV using PicoTip emitters (Waters Corporation) while

other source parameters (sampling cone 40 V; source off set 80 V; source temperature 800 °C; cone gas 50 l/h; nano gas flow 0.4 bar; and no purge gas) were not changed. The IMS was optimized for wave velocity by ramping with start velocity of 870 m/s to end at 564 m/s that corresponds to separation of GluFib fragments in the drift time range of 0-200 bins. The data acquisition was set up using the program MassLynx™ Software Version V4.1 (Waters Corporation) and it automatically switches between MS and MS/MS (HDMSE) scans that set at scan range 50–2000 m/z were acquired in resolution mode at 20000 with 1 sec scan time. GluFib was injected at an interval of 1 min and was used for calibration. The acquired data was analyzed for protein identifications using the program PLGS v3.3 (Water Corporation) against the FASTA database from Uniprot of V. vulnificus CMCP6 strain that contained 4417 sequences, 783 in Swissprot and 3684 in TrEMBL. For spectral processing, low and high energy thresholds of 135, 20 counts and lock mass calibration 785.8456 m/z for GluFib were used. The workflow search parameters contain trypsin as protease, one missed cleavage, carbamidomethyl for cysteine as fixed and oxidation of methionine as variable modifier. The protein quantification was carried out based on the top 3 peptides that had no modifications, pass one match having peptide fragment one and ranked first three highest peptides. The independent identification output by PLGS imported in ISOQuant 1.8[54] for comparison among all samples and quantification was done on proteins that were identified with minimum of two peptides and the protein areas used for subsequent analyses.

**Statistics and Reproducibility**. The data analysis was performed in R version 4.0.2 using the tidyverse package (version 1.3.0)[55]. Briefly, the IsoQuant protein intensities were median normalized using the global median as reference. The PCA analysis was carried out using the FactoMineR package (version 2.3)[56] with normalized log$_2$ protein intensities scaled to unit variance. The sample correlation was calculated using Kendalls methodology[57] and displayed using the ggcorrplot package (version 0.1.3)[58]. The statistical analysis was carried out using the PECA package (version 1.24.0)[59] by applying a modified t-test for the pairwise comparisons with proteins having valid protein quantity values in at least 2 replicates, by calculating an empirical Bayes moderated t-statistics using the linear modeling approach implemented in the limma package (version 3.44.3)[60]. The raw p-values (p) were multiple test adjusted (p.fdr) using the Benjamini-Hochberg method[61]. Volcano plots were generated using ggplot2 package (version 3.3.2)[62] with an absolute fold-change cutoff of 1.5 and 0.05 as q-value (adjusted p-value) cutoff. Venn diagrams were drawn using the tool available at: http://bioinformatics.psb.ugent.be/beg/software.

**Electron microscopy**. For negative stain electron microscopy, 3 µL of the sample were loaded on a freshly glow-discharged, carbon coated grid (400 mesh, SPI Supplies / Structure Probe, Inc). After incubation for 30 sec, excess liquid was blotted away and the sample was stained with 1% Uranyl acetate solution and imaged on a Philips CM120 electron microscope, operated at 120 kV. For vitrification, 3 µL of the purified, oxidized VvRsbRS complex at a concentration of 0.3 mg/mL were applied to glow-discharged Quantifoil holey carbon grids (Quantifoil Micro Tools). The sample was plunge frozen after blotting for 2.5 sec in a Vitrobot at 70% humidity and 10 °C. Grids were immediately transferred to and stored in liquid nitrogen. Cryo-EM data collection was performed manually over multiple sessions on a Tecnai Polara (Thermo Fisher Scientific) cryo-electron microscope (operated at 300 keV), equipped with a Falcon II detector (Thermo Fisher Scientific). Per exposure, 24 subframes were recorded with an exposure time of 1.5 s and an electron dose of 3 e$^-$/Å$^2$ per subframe. The target defocus range was −1.7 to −5.5 µm. The dataset was recorded at a magnification of ×59,000, resulting in a pixel size of 1.77 Å/px on the sample level.

Unless stated differently, single particle cryo-EM data analysis was performed using RELION-1.4 and RELION-2[63]. Global motion correction was performed in MotionCor[64]. CTF was estimated using CTFFIND4.1[65]. Autopicking yielded a total of 230,784 particles. After multiple rounds of 2D classification and thorough visual inspection of the remaining particles, a subset of 35,647 particles was used for further analyses. Initial models were created by RELION from the data themselves, or obtained from published structures (EMDB-ID 1555). A first 3D refinement was calculated applying icosahedral symmetry. Relion_project was used to calculate random projections from the 3D reconstruction; subsequently, 2D classification was used to obtain a more robust visualization of the most prominent views. A comparison of VvRsbRS 2D class averages and the class averages of the projections revealed significant inconsistencies. Icosahedral symmetry was therefore excluded for VvRsbRS. Eigenimages were calculated from a random subset of particles by SPIDER – PCA using 25 Eigenvectors[66], following the Scipion MDA Workflow[67]. The ten first Eigenimages revealed the presence of 2fold, 3fold and 5fold symmetry in the dataset. Consequently, ten initial models representing different variations of these symmetry groups were created. For supervised multi-reference 3D classification (using 10 classes and C1 symmetry), a.star file[38] pointing to the locations of the initial models was provided as a reference. The resulting distribution of particles in the ten 3D classes showed a strong preference for twofold symmetric classes; higher-symmetry classes were not populated. The final 3D reconstructions were therefore performed applying D2 point group symmetry. Subsequently, to improve the alignment of the STAS-domain core, the regions

containing the sensory domains were masked out and a reconstruction of the core was obtained.

**Homology modeling of the *Vv*RsbR2 and *Vv*RsbS2 dimers and model building of the *Vv*RsbR:*Vv*RsbS stressosome complex.** Homology detection was performed by HHblits[68] using the query sequences of *Vv*RsbR and *Vv*RsbS separately. The HMM database pdb70 was chosen to search and build multiple query-template alignments. The secondary structure of the query was predicted by PSIPRED and subsequently compared to the actual secondary structure of the database templates. The secondary structure similarity score was used to enhance the alignment quality. A local alignment mode was used to identify remote homologs sharing only a common core, revealing the query and template belonged to the same protein superfamily. Concerning the *Vv*RsbS STAS domain (made up of 115 residues), the HHblits search yielded the RSBS anti-sigma-factor antagonist from *Moorella thermoacetica* (pdb id 3ZXN, DOI:10.2210/pdb3zxn/pdb)[69] as the best candidate to generate a suitable homology model (sequence identity 25%). It was selected due to its high experimental resolution (1.9 Å). The secondary structure score was 14.9, the aligned columns were 115 (no gaps), and the sequence similarity was 0.525. The template sequence is 123 amino acids long, whose first 115 have been used in the alignment with the target. The homologous search of the *Vv*RsbR STAS domain (113 residues) led to the same template detection, suggesting 3ZXN as the most promising one (sequence identity 22%). The secondary structure score was 13.6, the aligned columns were 111 (with initial and final gaps) and the sequence similarity was 0.425. Residues 7 to 117 were used for alignment with the target sequence. Both *Vv*RsbR and *Vv*RsbS were assembled as dimers, in accordance to the template and to the map, using the CHIMERA program. The *Vv*RsbR N-terminal domain (composed of 138 amino acids) was modeled using the globin coupled sensor from *Geobacter sulfurreducens* (pdb id 2W31, DOI: 10.2210/pdb2w31/pdb)[70] as a template, with an experimental resolution of 1.5Å[70]. It shows a sequence identity of 21%, secondary structure score of 118.0 and a sequence similarity of 0.392 with 138 aligned columns (no gaps). The template sequence has a length of 162 amino acids, and those in the range from 13 to 150 were used in the alignment with the target sequence. The linker helix connecting the *Vv*RsbR N-terminus with the C-terminal STAS domain was modeled as a poly-Alanine helix made up of 20 amino acids. It was built using the COOT program[71] and fitted as a dimer into the density map using the FLEX-EM[72] method of the MODELLER program[73]. The three components of *Vv*RsbR (N-terminal sensor domain, linker helix and STAS domain) were assembled in a final, complete *Vv*RsbR2 dimer, fitted into the cryo-EM density map and refined by using phenix.real_space_refine without imposing constraints[74]. The *Vv*RsbS dimer was also fitted into the map. Fitting was performed initially by CHIMERA[75] and then adjusted by FLEX-EM. Copies of the *Vv*RsbR2 and *Vv*RsbS2 dimers were generated and fitted into the asymmetric cryo-EM density map. These global fittings were initially guided by CHIMERA and then refined by phenix.real_space_refine[74].

**Reporting summary**. Further information on research design is available in the Nature Research Reporting Summary linked to this article.

## Data availability

Proteomic data that support the findings of this study are available at MASSive with accession: MSV000087636. The structural data presented in this manuscript has been deposited in the wwPDB with the ID 7O01. The corresponding EM map can be found in the EMDB under the code EMD-12676. Availability of source data is as follows: Fig. 1a, b and supplementary Fig. 5b (Supplementary data 1), proteomic data shown in Supplementary Fig. 5c (Supplementary Data 2) and Fig. 5d (Supplementary Data 3 and 4). The occurrence of stressosome gene clusters in the genus *Vibrio* is summarized in supplementary data 5. All other data are available from the corresponding author on reasonable request. The uncropped blots for Figs. 1d and 2d are presented in Supplementary Fig. 18.

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

## Acknowledgements

This project has received funding from the European Union's Horizon 2020 research and innovation program under the Marie Skłodowska-Curie grant agreement No 721456 to AB, JM-W, RJL, CZ and JP-F, and was supported by a starting grant of the University of Greifswald to JP-F. RJL and JM-W thank the UK BBRSC for funding (BB/G001553/1). We thank Alejandro F. Alice and Jorge H. Crosa (Oregon Health & Science University, Portland) for kindly providing *V. vulnificus* CMCP6. We also thank Christoph Ruess for help with experiments and Michael Hecker and Susanne Engelmann for continuous support and fruitful discussions. Genome sequencing was provided by MicrobesNG (http://www.microbesng.uk) which is supported by the BBSRC (grant number BB/L024209/1).

## Author contributions

A.B., R.J.L., J.M.-W., J.P.-F., and C.Z., designed this study and the experiments. Performed experiments, analyzed and visualized the data - *V. vulnificus* experiments: M.C., L.C., A.G., P.B., and J.P.-F.; stressosome biochemistry and cryo-EM: G.B., V.H., W.J., S.K., J.R.L., M.G.M., J.M.-W., J.P.-F., and C.Z.; proteome analysis: K.R., V.M.D., M.L., S.M., J.P.-F., A.R., and U.V.; drafted and edited manuscript V.H., G.B., R.J.L., J.M.-W., J.P.-F., and C.Z. All authors discussed the results and commented on the manuscript.

## Funding

## Competing interests
