## [Peer Review File · Communications Biology]

Reviewers' comments:

Reviewer #1 (Remarks to the Author):

Heinz et al. have studied the Gram negative bacteria *Vibrio vulnificus* and provide new genetic, biochemical, in vivo and cryo electronmicroscopy structural data suggesting the involvement of the stressosome in sensing oxygen levels and role in iron metabolism. The data are interesting and convincing and as such will have an impact in the field. I have some comments that needs attention.

Page 4 In the introduction it would be informative to discuss and mention the aim and reason of this study.

Line 169 a minor revision change ...(reduced) (fig. 2c). to ...(reduced; fig. 2c).

Page 8 You mention fig. 5a, fig. 5b and fig. 5c I think you mean fig. S5a etc.

Line 192-197 this sentence is unclear and needs to be revised.

In the Discussion I missed a discussion that relates to mycobacteria, see for example Pané-Farré et al. 2005.

Line 477 minor revision in vitro should be in italic.

Line 545 minor revision insert comma before respectively.

Line 577 ...), a .star file ... this needs to be clarified/revised.

References, there are missing information for the following references 5, 19, 21, 22, 23, 24, 26, 31, 32, 35, 36, 38, 39, 40, 43, 47, 54, 55, 56, 57 and 58.

Main figures

Fig 1a error bars are missing.

Fig 2 line 820 Reference 165?

Supplementary information

Fig S1 Here I missed references that needs to be added.

Fig S4 line 95 ...the helix ??2 needs attention.

Fig S5a error bars are missing.

Fig S6 in the legend you state "Legend indicating sampling..." I cannot find this or see this, something is missing.

Supplementary Materials and Methods

Lines 242 and 245 *V. vulnificus* and *E. coli* should be in italic.

Line 253 add space between μg and ml^{-1} .

Line 284 for continuity throughout the manuscript either use 4°C or 4°C .

Line 319 MgCl_2 should read MgCl_2 .

Line 338 add space between 170 and min.

Line 340 clarify 800C.

Line 346 add space between 1 and sec.

Missing information for following supplementary references 4, 5, 11, 12 and 15.

Reviewer #2 (Remarks to the Author):

In this study, Heinz et al. perform an elegant and exhaustive analysis focused on the biology of the *Vibrio vulnificus* stressosome. They combine convincingly a large number of approaches, from bacterial physiology using appropriate mutants to phylogenomics, proteomics, two-hybrid assays and a in deep structural characterization of stressosome particles obtained from recombinant *E. coli* expressing VvRsbR and VvRsbS. The study provides a bulk a data that certainly would be valuable for future investigations in this field.

Comments

1) Figure 1 shows a key experiment in which interaction of VvRsbR and VvRsbS is shown by pull-down assays in the natural host, *V. vulnificus*. The R-S complexes that were analysed at the structural level, resulting in a stoichiometry of 2.4:1, are however obtained in recombinant *E. coli*. What was the difference in protein levels between one and the other system? Do the authors see feasible to obtain complexes from the natural host to perform comparable analyses? Do they see any caveat in producing the proteins in an heterologous system? These questions take into account

that the R:S ratio in natural host (Fig. 1, panel C) seems to be higher than 2.4:1. This could support a pool of free R protein if the 2.4:1 stoichiometry that is presumed to be conserved in *V. vulnificus*.

2) Fig. 1, panel B needs a loading control. This Western is important to convince on the regulation of R depending on the growth phase. Since anti-S antibodies are available, monitoring of S levels along the curve would be also of value. About panel C, an image of R and S levels in the input is needed. This pull-down assay should have been also performed with the null mutant as negative control to shown specificity of antibodies used.

3) Proteomics is a great approach to get a global picture of global physiological changes occurring. However, in some instances there are disparities with the real changes due to intrinsic properties of the protein, essentially amino acid composition, that may results in very long- short-peptides after trypsin digestion or peptides that ionize poorly. Since anti-R and anti-S antibodies are available, authors could test correspondence between proteomic data shown in Fig S5, panel b, with the respective Western assays in wild type and mutants backgrounds.

4) An analysis similar to that shown in Figure 3 could be depicted exclusively within the *Vibrio* genus. Although the authors do not discuss about that, it is quite relevant that *Vibrio cholerae*, also colonizing water habitats with different salt concentrations, it appears to a stressosome-based system. A major difference is that *V. cholerae* does not cause systemic disease, so control of iron metabolism might not be so critical. Indeed, *V. cholerae* pathology is essentially restricted to major inflammation in the intestinal epithelium. Can the authors speculate a bit on this?

5) In some instances, the text refers to "lower" or "higher" changes without specifying the exact data obtained, i.e. the fold change registered. This is important and need to be corrected. Some examples without providing fold-change or degree of change:

- L117: lower optical density...
- L195: little overlap...
- L214: higher level..
- L219: lower level...

6) Figure 2c is cited before 2b.

7) Lines 184-189: citation to figures 5a, 5b and 5c are incorrect. This was confusing. They are supplementary.

8) Fig. S6 has too small lettering. May be split in different panels?

9) L179: wild type (separate)

10) L326: Miksys et al. submission should be properly referenced.

11) L423: re-occurring? Replace by recurrent.

12) L427-431: indicate final pH of the medium at which it was adjusted.

13) L436: this hypoxia conditions ("filled Falcon tubes") seem hard to be reproduced by other labs. Did the authors attempt to use micro-anaerobic chambers using commercial reagents.

Reviewer #3 (Remarks to the Author):

This ms brings a valuable contribution to the understanding of the stressosome structure and function in an important gram-negative, pathogenic bacterium. The ms is well written and the data are sound, representing a significative step towards a deeper knowledge of these systems. I have just some observations/suggestions, that could be useful to improve the ms and render it more

clear for non specialists of the field:

-pg. 3-4 specify the names of the products of the eight and the four genes, stressing which proteins are missing in the latter instance;

-pg. 6, it is not clear if there is experimental evidence of a VvRsbT binding to the minimal stressosome formed with VvRsBR and VvRsBS. This point needs to be better clarified in view of the docking simulations; similar observations hold for the supposed interaction of RsbT with downstream partners (pg. 11);

-pg. 16: the possible link with stressosome, iron levels in the blood and severity of infection is too vague, needs a better perspective;

-lines 412-419: this paragraph is rather speculative, I would avoid it.

Point by point answers

We sincerely thank all three reviewers for their constructive comments: they certainly helped to improve the manuscript. Changes to the revised manuscript can be followed as “tracked changes” and are also included in the response letter and indicated by quotation marks. We have also added Supplementary Figures to our response letter to strengthen our line of argument. As a consequence of these changes, we have had to include Dr. Patricia Bedrunka, Philipps University of Marburg, as a new co-author. Patricia contributed substantially to the revision of the manuscript with the addition of new data shown in the main manuscript this response letter (**figure 1 main manuscript, figures 1 & 2 response letter**) and new text mostly on page 5.

Please find below point-by-point responses to all comments raised. The reviewers' comment is shown in black italic, while our answers are in blue.

Reviewer #1 (Remarks to the Author):

Main text

C1. *Page 4 In the introduction it would be informative to discuss and mention the aim and reason of this study.*

A1. We included the sentence in the introduction: “The aim of this study was to explore the structure and function of the thus far neglected stressosomes of Gram-negative bacteria.”

C2. *Line 169 a minor revision change (reduced) (fig. 2c). to (reduced; fig. 2c).*

A2. Changed as suggested.

C3. *Page 8 You mention fig. 5a, fig. 5b and fig. 5c I think you mean fig. S5a etc.*

A3. Yes. Changed.

C4. *Line 192-197 this sentence is unclear and needs to be revised.*

A4. We changed the wording to: “The number of proteins with significantly changed abundance (1.5-fold and $p < 0.05$) in the $\Delta rsbRSTX$ versus wild type comparison was much smaller in exponentially growing cells than in stationary phase cells (**fig. 5Sc**). This suggests that a stationary phase signal, correlating with exhaustion of glucose, triggers a strong stressosome response. Following the hypoxic-shift, the lack of the stressosome caused a more pronounced down-regulation than upregulation of proteins in the stressosome mutant at all time points (**fig. 5Sc**), pointing to the importance of the stressosome to maintain protein expression in oxygen-restricted conditions. Although in all three

conditions – growth, stationary phase and hypoxia – the loss of the stressosome resulted in pronounced changes in protein expression, little overlap in the stressosome-dependent proteomic signatures was observed (**fig. S5d, tab. S4 and S5**).”

C5. *In the Discussion I missed a discussion that relates to mycobacteria, see for example Pané-Farré et al. 2005.*

A4. Work describing the presence of stressosomes in high-GC Gram-positive bacteria is now included in the discussion: “The role of the stressosome in high-GC Gram-positives and Gram-negatives is largely unexplored. Recent pioneering work in *Mycobacterium marinum* showed that expression and cellular localization of stressosomes varies in response to stress and growth conditions in this species^{1,2}.”

To further highlight that stressosomes are likely to play important roles in Gram-negatives, we also included a reference to the Gram-negative *Xanthomonas campestris*, where growth phase dependent expression and phosphorylation of the RsbR protein has been reported: “The *Vibrio basiliensis* stressosome complex is currently the only example studied from a Gram-negative bacterium. However, expression and phosphorylation of stressosome core protein XcRsbR was also reported in the Gram-negative plant pathogen *Xanthomonas campestris* during transition from the late exponential growth phase to the stationary phase^{3,4}.”

C6. *Line 477 minor revision in vitro should be in italic.*

A6. Changed as suggested.

C7. *Line 545 minor revision insert comma before respectively.*

A7. Changed as suggested.

C8. *Line 577, a .star file. this needs to be clarified/revised.*

A8. We included a reference for the star file format.

C9. *References, there are missing information for the following references 5, 19, 21, 22, 23, 24, 26, 31, 32, 35, 36, 38, 39, 40, 43, 47, 54, 55, 56, 57 and 58.*

A9. We thank the reviewer for thoroughly checking the references and we have now included the missing information.

Main figures

C10. *Fig 1a error bars are missing.*

A10. Figure 1a was changed to include error bars.

C11. Fig 2 line 820 Reference 165?

A11. The number 165 specifies the residues covered by the VvRsbR N-terminal construct. To make this more clearly the index 165 was deleted and replaced by “amino acid residues 1-165” in the text.

Supplementary information

C12. Fig S1 Here I missed references that needs to added.

A12. References are now provided in the legend of figure S1.

C13. Fig S4 line 95 the helix ??2 needs attention.

A13. Formatting error corrected.

C14. Fig S5a error bars are missing.

A14. Error bars showing standard deviation are now added to figure S5a. At low OD values the errors are too small to be visualized behind the graph symbols. Please see table below.

WT			ΔRSTX		
time in h	mean OD	SD of OD	time in h	mean OD	SD of OD
0	0.048	0.001	0	0.043	0.001
1	0.078	0.002	1	0.073	0.003
2	0.132	0.002	2	0.134	0.006
3	0.305	0.005	3	0.274	0.008
3.75	0.512	0.010	3.75	0.510	0.009
4	0.612	0.011	4	0.598	0.063
5	1.383	0.019	5	1.343	0.048
6	2.343	0.272	6	2.733	0.126
7	2.665	0.114	7	2.765	0.021
8	2.703	0.205	8	2.550	0.160
9	2.538	0.077	9	2.483	0.043

WT- O2			ΔRSTX - O2		
time in h	mean OD	SD of OD	time in h	mean OD	SD of OD
0	0.052	0.001	0	0.049	0.001
1	0.061	0.001	1	0.077	0.002
2	0.111	0.002	2	0.154	0.004
3	0.231	0.005	3	0.347	0.008
4	0.526	0.018	4	0.560	0.014
4.5	0.612	0.011	4.5	0.626	0.024
5	0.692	0.009	5	0.725	0.012
6	1.058	0.038	6	0.913	0.022

C15. Fig S6 in the legend you state "Legend indicating sampling ..." I cannot find this or see this, something is missing.

A15. This statement points to the caption of the graph in the lower left corner, which specifies the growth conditions for the individual samples. To make this point more clearly the statement was changed to: "The legend indicating growth conditions for the individual samples is shown below the left lower graph".

Supplementary Materials and Methods

C16. Lines 242 and 245 *V. vulnificus* and *E. coli* should be in italic.

A16. done

C17. Line 253 add space between 100 μg and ml^{-1} .

A17. done

C18. Line 284 for continuity throughout the manuscript either use 4°C or 4 °C.

A18. Done, changed to 4 °C

C19. Line 319 MgCl_2 should read MgCl_2 .

A19. done

C20. Line 338 add space between 170 and min.

A20. done

C21. Line 340 clarify 800C.

A21. done

C22. Line 346 add space between 1 and sec.

A22. done

C23. Missing information for following supplementary references 4, 5, 11, 12 and 15.

A23. We thank the reviewer for thoroughly checking the references and included the missing information.

Reviewer #2 (Remarks to the Author):

In this study, Heinz et al. perform an elegant and exhaustive analysis focused on the biology of the Vibrio vulnificus stressosome. They combine convincingly a large number of approaches, from bacterial physiology using appropriate mutants to phylogenomics, proteomics, two-hybrid assays and a in deep structural characterization of stressosome particles obtained from recombinant E. coli expressing VvRsbR and VvRsbS. The study provides a bulk a data that certainly would be valuable for future investigations in this field.

We thank reviewer #2 for the positive and constructive feedback.

Comments

C1. *Figure 1 shows a key experiment in which interaction of VvRsbR and VvRsbS is shown by pull-down assays in the natural host, V. vulnificus. The R-S complexes that were analysed at the structural level, resulting in a stoichiometry of 2.4:1, are however obtained in recombinant E. coli. What was the difference in protein levels between one and the other system?*

A1.1 We did not compare the VvRsbR to VvRsbS ratio during natural expression in *V. vulnificus* and heterologous expression in *E. coli*: due to the low abundance of VvRsbS in *V. vulnificus*, we were unable to directly quantify VvRsbS (by proteomics or by Western blotting) in extracts of *V. vulnificus*. VvRsbS was detectable in Western blot experiments only when stressosomes were enriched by immunoprecipitation.

Do the authors see feasible to obtain complexes from the natural host to perform comparable analyses? Do they see any caveat in producing the proteins in an heterologous system? These questions take into account that the R:S ratio in natural host (Fig. 1, panel C) seems to be higher than 2.4:1. This could support a pool of free R protein if the 2.4:1 stoichiometry that is presumed to be conserved in V. vulnificus.

A1.2 This is indeed an important aspect brought up by the reviewer. We agree that stressosomes purified from *V. vulnificus* would be important to clarify the VvRsbR to VvRsbS stoichiometry *in vivo*, and whether alternative stoichiometries - as suggested by the analysis of recombinant stressosomes - coexist in the natural setting. Comparing VvRsbR levels from purified stressosome particles with the VvRsbR level in whole *V. vulnificus* cell extracts could reveal if a pool of free VvRsbR might exist. However, due to the technical challenges associated with the purification of stressosomes from *V. vulnificus* in quantities sufficient for a CryoEM analysis, we did not aim to conduct such an analysis.

Therefore, to emphasize the important aspect brought up by the reviewer, the following sentence was included in the discussion: “It should be considered here, however, that the minimal VvRsbR:VvRsbS complex investigated in this study has been heterologously expressed and purified from *E. coli* (which is also the case for all other published stressosome structures). It is thus possible that the stoichiometry adopted by native stressosome complexes is different to these recombinant stressosomes.”

C2. Fig. 1, panel B needs a loading control. This Western is important to convince on the regulation of *R* depending on the growth phase. Since anti-*S* antibodies are available, monitoring of *S* levels along the curve would be also of value.

A2.1 We acknowledge the point of the reviewer. We repeated the Western blot experiment and compared VvRsbR levels during exponential growth and four hours after entry into stationary phase of wild-type *V. vulnificus* (CMCP6) cells cultured in medium with and without iron supplement. We also tested an isogenic *V. vulnificus* mutant lacking the *rsbRSTX* gene cluster (CMCP6 Δ RSTX) as control for antibody specificity. As a control for loading and quantitative blotting of proteins, the Western blot membranes were stained with Ponceau red prior to immunodetection.

The Western blot analysis clearly confirmed the stationary phase induction of VvRsbR (**response letter figure 1, main manuscript figure 1**). Several attempts to detect VvRsbS were unsuccessful, probably due to its low abundance.

Figure 1. VvRsbR accumulation during growth. Panel A shows the raw data for the growth phase (e= exponential, s = stationary) dependent accumulation of VvRsbR (arrowhead) in medium with (+FeCl₃) or without (-FeCl₃) iron supplement. Top row shows Ponceau red stained membranes after protein transfer. Bottom row shows the corresponding Western blot detection of VvRsbR. Molecular size marker is indicated at the left side. Panel B shows quantified signal intensities of the three biological

replicates shown in A. Error bars are standard deviations. Significance analysis by Students T-test (two sided, paired samples): * $p < 0.05$, ** $p < 0.01$.

About panel C, an image of R and S levels in the input is needed. This pull-down assay should have been also performed with the null mutant as negative control to shown specificity of antibodies used.

A2.2 This is a valid concern. Specificity of the anti-VvRsbR antibody is addressed in the experiment shown above. Because VvRsbS is not detectable in *V. vulnificus* without prior enrichment of stressosome complexes, the specificity of the VvRsbS antibody was validated using an *E. coli* strain expressing the VvRsbR:VvRsbS core complex from a single plasmid under control of an IPTG-inducible promoter. VvRsbS expression was detected by immunoblotting before and after addition of IPTG to the *E. coli* culture (**response letter Figure 2**). The Western blot analysis showed a clear increase in a signal at an apparent molecular mass that corresponds to VvRsbS in the induced samples.

Figure 2. anti-VvRsbS antibody test. Panel A shows SDS PAGE of protein extracts from *E. coli* BL21 DE3 cells expressing VvRsbR and VvRsbS under control of an IPTG-inducible promoter. (-) no addition of IPTG, (+) 3h after induction. The asterisks mark the expected positions by mass of VvRsbR and VvRsbS. Panel B shows the corresponding Western blot experiment probed with the anti-VvRsbS antibody. To show that the VvRsbR and VvRsbS are soluble during co-expression the cleared cell lysate was analyzed together the extract of whole cells without prior removal of cell debris and insoluble material by centrifugation.

C3. Proteomics is a great approach to get a global picture of global physiological changes occurring. However, in some instances there are disparities with the real changes due to intrinsic properties of the protein, essentially amino acid composition, that may result in very long- short-peptides after trypsin digestion or peptides that ionize poorly. Since anti-R and anti-S antibodies are available,

authors could test correspondence between proteomic data shown in Fig S5, panel b, with the respective Western assays in wild type and mutants backgrounds.

A3. We absolutely agree with the reviewer that quantitative proteomics is affected by the size and the capacity to ionize peptides in the sample. Although independent confirmation of the proteomic results would be desirable, we were unable to detect VvRsbS directly in *V. vulnificus* extracts. VvRsbS was only detected after enrichment of the complex by immunoprecipitation. However, we were able to test correspondence of our proteomic data and the Western blot results for VvRsbR. The proteomic data suggested an approximately 6-fold increase of VvRsbR upon stationary phase induction (**supplementary Figure S5B**). When probed by Western blot analysis, values around 4-fold were calculated for the same protein samples (**response letter Figure 1**). Some of the protein samples from the Δ RSTX mutant during exponential growth display a weak signal at a molecular mass close to where the VvRsbR signal appears. Therefore, the calculated VvRsbR intensities for the exponential phase samples might be an overestimate of the actual VvRsbR amount suggesting even more than 4-fold induction. Thus, both methods show a similar magnitude of regulation and support the conclusion that VvRsbR is induced upon stationary phase entry.

C4. *An analysis similar to that shown in Figure 3 could be depicted exclusively within the Vibrio genus. Although the authors do not discuss about that, it is quite relevant that Vibrio cholerae, also colonizing water habitats with different salt concentrations, it appears to a stressosome-based system. A major difference is that V. cholerae does not cause systemic disease, so control of iron metabolism might not be so critical. Indeed, V. cholerae pathology is essentially restricted to major inflammation in the intestinal epithelium. Can the authors speculate a bit on this?*

A4. We agree with the reviewer and have updated the manuscript with an analysis of the occurrence of stressosomes within the genus *Vibrio*. A supplementary table (table S6) summarizing this analysis is now included in the manuscript and the search strategy for the bioinformatic analysis described in the supplementary materials. The currently available information does not suggest an obvious explanation for the observed distribution of stressosomes among *Vibrio* species: e.g. why it is not present in *V. cholerae* (e.g. no clear habitat pattern, no high phylogenetic affinity among species encoding stressosomes etc). Therefore, to summarize this observation and analysis, the following sentence was included in the results: "A screen of complete proteomes available at UniProt for the genus *Vibrio* identified the complete set of stressosome proteins (RsbR, RsbS, RsbT and RsbX) in a surprisingly small number of *Vibrio* species (four out of 38; tab. S6): *V. mangrove*, *V. nigripulchritudo*, *V. pectenicida* and *V. vulnificus*. These species all share a similar set of potential downstream regulators. Why stressosomes are restricted to only a few *Vibrio* species, and if this is a common phenomenon in other genera, is presently unknown."

C5. *In some instances, the text refers to lower or higher changes without specifying the exact data obtained, i.e. the fold change registered. This is important and need to be corrected. Some examples without providing fold-change or degree of change:*

- L117: *lower optical density; corrected*
- L195: *little overlap; please see below*
- L214: *higher level; corrected*
- L219: *lower level; corrected*

A5. The manuscript was checked for further indication of condition-dependent changes and values were added where missing. We also modified and extended the paragraph addressing proteomic changes in the abundance of translation related proteins (please see below). However, we did not include numbers for the overlaps of the individual proteomic signatures, which can be taken from figure S5d. Including these numbers would decrease substantially the readability of the text.

“Moreover, a consistent upregulation in the stressosome mutant was observed for an uncharacterized ABC-F protein (VV1_0491, between 25 to 29-fold) showing homology to the translation throttle Etta⁵. Additional proteins with a role in translation were differentially regulated in the mutant compared to the wild type. In stationary phase cells of the stressosome mutant we observed upregulation of the ribosome hibernation protein (VV1_0693, 4.7-fold), a peptidyl-tRNA hydrolase (VV1_0258, 4.3-fold) and proteins involved in tRNA biogenesis and modification (VV1_0266, 2.2-fold; VV1_0277, 2.1-fold). Finally, a potential translation release factor methyltransferase (VV1_0252, 2.9-fold) and several proteins with a role in tRNA modification (VV1_1251, VV1_2142, VV1_2608; VV1_2926) were present at lower level (1.5 to 1.9-fold) in the mutant, pointing to a potential impact of stressosome activity in modifying translation in *V. vulnificus* (tab. S3).”

5. Boël, G. et al. The ABC-F protein ETTA gates ribosome entry into the translation elongation cycle. *Nat. Struct. Mol. Biol.* 21, 143–51 (2014).

C6. *Figure 2c is cited before 2b.*

A6. Changed.

C7. *Lines 184-189: citation to figures 5a, 5b and 5c are incorrect. This was confusing. They are supplementary.*

A7. We apologize for this mistake. The citation to the figures is now corrected.

C8. *Fig. S6 has too small lettering. May be split in different panels?*

A8. Done as suggested.

C9. L179: *wild type (separate)*

A9. done

C10. L326: *Miksys et al. submission should be properly referenced.*

A10. Changed to: Miksys et al., in revision, COMMSBIO-21-1365A

C11. L423: *re-occurring? Replace by recurrent.*

A11 done

C12. L427-431: *indicate final pH of the medium at which it was adjusted.*

A12. done

C13. L436: *this hypoxia conditions (filled Falcon) seem hard to be reproduced by other labs. Did the authors attempt to use micro-anaerobic chambers using commercial reagents?*

A13. We appreciate the reviewer's comment. It was also our concern to establish reproducible procedures for the induction of hypoxia. We did not use micro-anaerobic chambers because we successfully used induction of hypoxia by transfer of the culture to tightly sealed tubes already in previous experiments with *Staphylococcus aureus* (PMID: 33526614). Careful monitoring of oxygen levels in these experiments showed a complete consumption of the dissolved oxygen within several minutes and highly reproducible kinetics of oxygen depletion for *S. aureus*. Since growth of the *V. vulnificus* was very reproducible following the shift of the culture to the sealed tubes, we concluded that induction of hypoxia also follows a reproducible kinetic in the *V. vulnificus* experiment. Thus, giving the same temperature and culture volumes, we expect that other labs will be able to reproduce our hypoxia experiment.

Reviewer #3 (Remarks to the Author):

C1. *This ms brings a valuable contribution to the understanding of the stressosome structure and function in an important gram-negative, pathogenic bacterium. The ms is well written and the data are sound, representing a significant step towards a deeper knowledge of these systems. I have just some observations/suggestions, that could be useful to improve the ms and render it more clear for non specialists of the field:*

A1. Thank you for the positive feedback

C2. -pg. 3-4 specify the names of the products of the eight and the four genes, stressing which proteins are missing in the latter instance;

A2. Done, we have included a modified paragraph: “Phylogenetic analysis suggests that the extended eight-gene stressosome operon (*rsbRSTUVWsigBrsbX*) encoding the three stressosome proteins (RsbR, RsbS and RsbT), the stressosome feedback-phosphatase (RsbX), and the SigB activation cascade (RsbU, RsbV and RsbW) is not common in nature, and seems restricted to the order Bacillales. Instead, a four-gene module (*rsbRSTX*) encoding a minimal stressosome but lacking the *sigB* gene and the SigB activation cascade (*rsbU, rsbV and rsbW*), is widely conserved in diverse bacteria including Cyanobacteria, Bacteroidetes, Proteobacteria, Deinococci and even some archaeal species”.

C3. -pg. 6, it is not clear if there is experimental evidence of a VvRsbT binding to the minimal stressosome formed with VvRsbR and VvRsbS. This point needs to be better clarified in view of the docking simulations; similar observations hold for the supposed interaction of RsbT with downstream partners (pg. 11);

A3. To clarify why the BACTH assay failed to reveal interactions with either VvRsbR or VvRsbS we extended the statement in the manuscript emphasizing that the RsbR:RsbS complex is required to allow RsbT binding in *Bacillus*: “Lack of VvRsbT binding is in accord with results from the *B. subtilis* stressosome, where the binding of RsbT is dependent on the formation of the RsbR:RsbS complex⁶, a condition not captured by the BACTH assay. BACTH analyses were carried out in aerobic conditions, which might affect the binding of VvRsbT to VvRsbR.”

6. Chen, C. C., Lewis, R. J., Harris, R., Yudkin, M. D. & Delumeau, O. A supramolecular complex in the environmental stress signalling pathway of *Bacillus subtilis*. *Mol. Microbiol.* **49**, 1657–1669 (2003).

We made several attempts to reconstitute the ternary VvRsbR-VvRsbS-VvRsbT complex *in vitro* using recombinant co-purified VvRsbR-VvRsbS complex and purified VvRsbT protein. In gel filtration experiments a only a very small amount of VvRsbT coeluted with the VvRsbR:VvRsbS complex, even when VvRsbT was provided in excess.

The following sentence was included to indicate that experiments investigating the VvRsbT:VvD1 interaction were carried out: “Signaling by the stressosome in these cases could be transmitted through RsbT-dependent phosphorylation of the downstream sensor kinase, which would most likely be a transient interaction as required by a regulatory switch. This explanation is consistent with our

failure to produce positive interaction results for VvRsbT and VvD1, the two-component protein encoded directly down-stream of the stressosome module (data not shown).”

C4. -pg. 16: *the possible link with stressosome, iron levels in the blood and severity of infection is too vague, needs a better perspective;*

A3. We extended the paragraph discussing a potential link between the stressosome, iron metabolism and infection: “A prominent group of proteins linked to iron-metabolism and -uptake showed a consistent decrease in abundance in the stressosome mutant in all growth conditions tested. This finding could point to a reduced requirement for iron in the absence of an active stressosome or in a reduced capacity to induce iron-uptake pathways. At the very least the data suggest that the stressosome alters iron-metabolism in *V. vulnificus*; which could be relevant given the stressosome gene cluster is often associated with clinical isolates of *V. vulnificus*, as high blood iron is a risk factor for the development of severe *V. vulnificus* infection⁷”

C5. -lines 412-419: *this paragraph is rather speculative, I would avoid it.*

A3. The following paragraph was deleted from the manuscript: “In HemAT, oxygen binding by one subunit will likely decrease ligand-binding affinity of the second subunit as a consequence of the structural arrangement following the first binding event⁴⁰. As a result of this negative cooperativity, it could require several orders of magnitude of increasing O₂ concentrations to saturate the second subunit of HemAT. If this phenomenon was re-capitulated in the *V. vulnificus* stressosome it would produce a receptor system with a wide dynamic response range. Thus, the *V. vulnificus* stressosome could be particularly well suited to sense changes in oxygen availability that have to pass a certain threshold in order to initiate a full response.”

References

1. Pettersson, B. M. F. *et al.* Identification and expression of stressosomal proteins in *Mycobacterium marinum* under various growth and stress conditions. *FEMS Microbiol. Lett.* **342**, 98–105 (2013).
2. Ramesh, M. *et al.* Intracellular localization of the mycobacterial stressosome complex. *Sci. Rep.* **11**, 10060 (2021).
3. Jia, X., Wang, J., Rivera, S., Duong, D. & Weinert, E. E. An O₂-sensing stressosome from a Gram-negative bacterium. *Nat. Commun.* **7**, 12381 (2016).
4. Musa, Y. R. *et al.* Dynamic protein phosphorylation during the growth of *Xanthomonas campestris* pv. *campestris* B100 revealed by a gel-based proteomics approach. *J. Biotechnol.*

167, 111–22 (2013).

5. Boël, G. *et al.* The ABC-F protein EttA gates ribosome entry into the translation elongation cycle. *Nat. Struct. Mol. Biol.* **21**, 143–51 (2014).
6. Chen, C. C., Lewis, R. J., Harris, R., Yudkin, M. D. & Delumeau, O. A supramolecular complex in the environmental stress signalling pathway of *Bacillus subtilis*. *Mol. Microbiol.* **49**, 1657–1669 (2003).
7. Oliver, J. D. The Biology of *Vibrio vulnificus*. *Microbiol. Spectr.* **3**, (2015).

REVIEWERS' COMMENTS:

Reviewer #1 (Remarks to the Author):

This interesting study by Heinz et al. have been revised and my comments have been addressed. This work will be a valuable contribution to our understanding of the stressosome in bacteria, in particular with respect to Gram negative bacteria. I do have some very minor text revision that needs attention.

Page 11 line 265 delete the space aftermodule.

Page 15 line 357 delete the second "...is not pronounced,....".

Page 16 line 372 the reference 38 refers to the Xanthomonas study I think you intended to have reference 52 here, renumber the references accordingly.

Page 19 line 451 delete the space afteranalyses.

Page 23 line 557 delete the space after ...2163.

Page 24 lines 558 and 561 change 37oC to 37 °C.

Page 37 line 867 delete or clarify XXX.

Page 39 line 926 insert , after background.

Supplementary

Page 8 line 125 change (C) to (c).

Page 21 line 361 *Vibrio vulnificus* should be in italic.

Reviewer #2 (Remarks to the Author):

The authors have improved the manuscript and convincingly addressed my comments. It is certainly a nice piece of work combining different techniques, reaching conclusions supported by the data shown and providing valuable insights into the biological role of the stressosome in Gram-negative bacteria, for which little was known.

Reviewer #3 (Remarks to the Author):

All comments and suggestions have been addressed. The ms is ready to be published

Point by point answers

We sincerely thank all three reviewers for taking the time to review our manuscript and for their constructive comments: they certainly helped to improve the manuscript.

Reviewer #1 (Remarks to the Author):

This interesting study by Heinz et al. have been revised and my comments have been addressed This work will a valuable contribution to our understanding of the stressosome in bacteria, in particular with respect to Gram negative bacteria. I do have some very minor text revision that needs attention.

Main

Page 11 line 265 delete the space after "module".

done

Page 15 line 357 delete the second ".is not pronounced,".

done

Page 16 line 372 the reference 38 refers to the Xanthomonas study I think you intended to have reference 52 here, renumber the references accordingly.

Yes, reference is now corrected.

Page 19 line 451 delete the space after "analyses".

done

Page 23 line 557 delete the space after "2163".

done

Page 24 lines 558 and 561 change 37oC to 37 °C.

done

Page 37 line 867 delete or clarify XXX.

done

Page 39 line 926 insert , after background.

done

Supplementary

Page 8 line 125 change (C) to (c).

done

Page 21 line 361 *Vibrio vulnificus* should be in italic.

done

Reviewer #2 (Remarks to the Author):

The authors have improved the manuscript and convincingly addressed my comments. It is certainly a nice piece of work combining different techniques, reaching conclusions supported by the data shown and providing valuable insights into the biological role of the stressosome in Gram-negative bacteria, for which little was known.

Reviewer #3 (Remarks to the Author):

All comments and suggestions have been addressed. The ms is ready to be published.